# Stable oxygen isotopes of crocodilian tooth enamel allow tracking Plio-Pleistocene evolution of freshwater environments and climate in the Shungura Formation (Turkana Depression, Ethiopia).

Axelle Gardin[1], Emmanuelle Pucéat[2], Géraldine Garcia[1], Jean-Renaud Boisserie[1, 3], Adélaïde Euriat[1], Michael M. Joachimski[4], Alexis Nutz[5], Mathieu Schuster[6], Olga Otero[1]

[1]PALEVOPRIM, UMR 7262 CNRS, Université de Poitiers, Poitiers, France
[2]Biogéosciences, UMR 5561 CNRS, Université de Bourgogne, Dijon, France
[3]Centre Français des Études Éthiopiennes, USR 3137 CNRS & Ministère des Affaires Etrangères, Addis Ababa, Ethiopia
[4]GeoZentrum Nordbayern, Friedrich-Alexander Universität Erlangen-Nürnberg, Erlangen, Germany
[5]CEREGE, Aix-Marseille Université, CNRS, IRD, Collège de France, INRAE, Aix-en-Provence, France
[6]Université de Strasbourg, CNRS, Institut Terre & Environnement de Strasbourg, UMR 7063, Strasbourg, France

*Correspondence to*: Axelle Gardin (axelle.gardin@univ-poitiers.fr)

**Abstract.**

This study adopts a new approach describing paleohydrology and paleoclimates based on the interpretation of stable oxygen isotopes ($\delta^{18}O_p$) recorded in fossil crocodilian teeth. They represent an archive of prime interest for tracking freshwater paleoenvironmental change, applicable for many paleontological localities in the world: crocodilian teeth are abundant in continental basins and widely distributed since their diversification during the Mesozoic; the enamel phosphate is resistant to diagenesis and retains its original isotopic composition over geological timescales; their $\delta^{18}O_p$ mainly relies on that of the crocodilian's home water body ($\delta^{18}O_w$), which in turn reflects water body types, regional climate, and evaporation conditions. This study presents the first application of this theoretical interpretative model to the Shungura Formation (Lower Omo Valley, Ethiopia), a key witness of the important environmental change in eastern Africa during the Plio-Pleistocene that impacted the evolution of regional faunas, including humans. In this complex and variable environmental context, the $\delta^{18}O_p$ of coexisting crocodilians allows for fingerprinting the diversity of aquatic environments they had access to at a local scale. This study sheds light on two important results: the $\delta^{18}O_p$ of crocodilian teeth indicates (1) stable aquatic environments in the northern Turkana Depression from 2.97 Ma to ca. 2.57 Ma, but a decline in local waterbodies diversity after 2.32 Ma suggests increasing aridity and, (2) like previous geochemical studies on paleosols and bivalves in the area, show a significant increase in $\delta^{18}O_w$ from 2.97 Ma to ca. 1.14 Ma, likely due to the shifting air streams convergence zones between the West African and Indian Summer Monsoons and/or reduced rainfall over the Ethiopian Highlands.

## 1 Introduction

Water availability is the main factor constraining the distribution of continental flora and fauna, in close connection to regional and local hydrography (climate, physiography of watersheds, geodynamics) (e.g., Bauder et al., 2005; Otero et al., 2011; Mushet et al., 2019; Smith and Boers, 2023). Freshwater environments, therefore, constitute vital components of regions that have yielded fossil hominid remains, but they are often overlooked and described in a more simplified way compared to terrestrial ones.

Over the last million years, in eastern Africa, the Turkana Depression experienced significant environmental change, particularly in terms of hydrology (e.g., aridification, landscape opening, lake level fluctuation), however new studies have recently challenged the previous models (e.g., level fluctuations of the paleolake that occupied the Turkana Depression in Nutz et al., 2017, 2020). This region may have hosted a great diversity of types of freshwater environments (hereafter also called water bodies, including channels, rivers, deltas, lakes, swamps, ponds, floodplains, etc., Fig. 1), whose spatial distribution and evolution across time remain to be studied (Haesaerts et al., 1983; Nutz et al., 2017, 2020). The Shungura Formation (Fm) in southwestern Ethiopia, located upstream of the delta of the Omo River flowing in Lake Turkana, indicates the continuous presence of water during the Plio-Pleistocene and is thus interpreted as a refuge area for wildlife (Bobe, 2006) (Fig. 1). Freshwater environments of the Shungura Fm have primarily been explored through sedimentological studies and the analysis of invertebrate assemblages (e.g., Van Damme et Gautier 1972; Peypouquet et al. 1979, 1980; Heinzelin et al. 1983; Van Boxclaer et Van Damme 2009). However, to date, only about 15% of studies related to the Shungura Fm have addressed the aquatic component of its ecosystems. This might be mainly due to the terrestrial ecology of humans, which has led to a

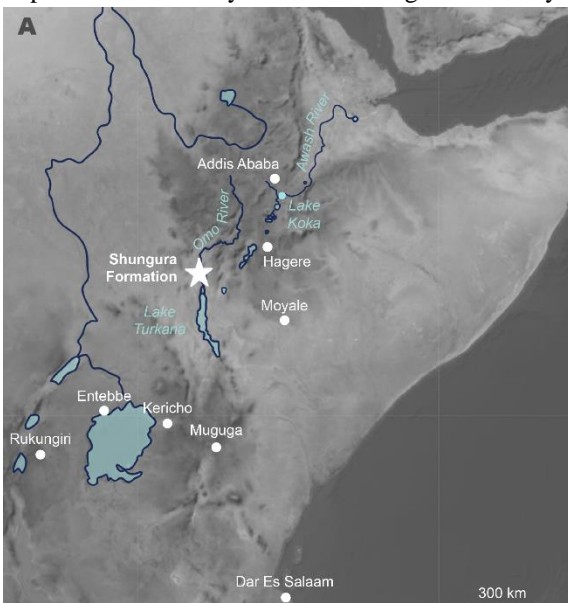
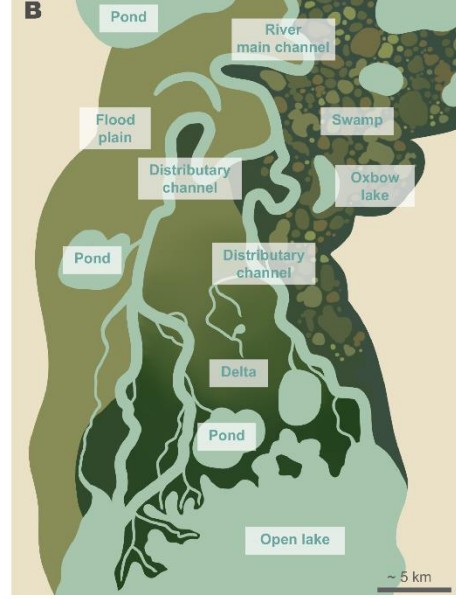

**Figure 1. A. Map of location and schematic representation of freshwater environments expected in a lacustrine delta: A, Map displaying the geographical position of the Shungura Formation and Lake Turkana in the Turkana Depression in eastern Africa and of GNIP stations used in the study; B, Diagram illustrating the theoretical diversity of freshwater environments and types of water bodies that may be present at a local scale in a lacustrine delta context, such as in the Turkana Depression.**

concentration on the terrestrial components of ecosystems. Nevertheless, it is essential to recognize the significance of aquatic environments in understanding the intricate interactions among climate, geodynamics, and aquatic communities. Therefore,

there is a lack of studies, tools, and methodologies suitable for describing aquatic environments with the same precision as terrestrial environments. This scarcity of focus on aquatic environments in these contexts makes it difficult to obtain comprehensive paleoenvironmental reconstructions.

In this context, the study of freshwater ectotherm vertebrates can provide original and valuable insights into the associated environmental conditions, such as hydrographic network connections, drainage characteristics, water body depth, salinity

levels, temperatures, and seasonal climatic shifts (e.g., Otero et al., 2011). Such features differ from the environmental information gained through the study of terrestrial fauna, which is highly dependent on its ecology.

The present study focuses on the oxygen isotopic composition of crocodilian teeth enamel phosphate ($\delta^{18}O_p$) to track hydrological change over time. Crocodilian enamel apatite is an ideal paleoenvironmental archive that can be used in multiple sites, as it is highly resistant to diagenetic and dissolution-precipitation processes (Kolodny et al., 1983, 1996; Shemesh et al.,

1983; Lécuyer et al., 1999) and found in abundance in continental sediments since the expansion of this clade in the Jurassic, especially in African continental series (e.g., Arambourg, 1967; Tchernov, 1976, 1986; Markwick, 1998; Brochu et al., 2022). As described hereafter, crocodilians have a low metabolism and are aquatic, this means that the interpretation of their $\delta^{18}O_p$ is much more constrained because it mainly reflects $\delta^{18}O$ of the water body ($\delta^{18}O_w$) they occupied during tooth formation (over several months). The linear relationship between $\delta^{18}O_p$ of crocodilian apatite and $\delta^{18}O_w$ (Amiot et al., 2007) enables to estimate

$\delta^{18}O_w$ from $\delta^{18}O_p$, thus providing valuable insights into aquatic environmental conditions. Previous analyses of $\delta^{18}O_p$ on crocodilians were systematically interpreted together with isotopic data from other sympatric vertebrates (fish, turtles, dinosaurs, mammals). This approach can help to infer the thermophysiology and ecology of taxa of interest (e.g., Amiot et al., 2010a), to estimate paleotemperatures (e.g., Amiot et al., 2008), aridity (Amiot et al., 2010b), to identify terrestrial or aquatic, freshwater or marine lifestyles (e.g., Goedert et al., 2020) or to differentiate the water sources used by different co-existing

taxa (Suarez et al. 2012).

This paper proposes a new theoretical interpretative model of (1) the range of $\delta^{18}O_p$ values recorded in crocodilian teeth to describe the diversity of waterbodies accessible to crocodilians at a local scale, as well as (2) the evolution of minimum $\delta^{18}O$ values over time, by using the comprehensive knowledge of the physiology and ecology of crocodilians. This approach is adapted and applied to fossil teeth from the Shungura Fm, representing an ideal framework for the first application, due to the

continuous sedimentary record from ca. 3.75 Ma and ca. 1.09 Ma (Boisserie et al., 2008; Table S1). The Shungura Fm reveals diverse aquatic environments, related to lake level fluctuations, mainly fed by two monsoons, despite a complex, highly variable, and increasingly arid context (Haesaerts et al. 1983; Kebede and Travi, 2012; Nutz et al. 2020). The growing knowledge of the evolution of its environments, the continuity of the series over several million years, and its precise dating allow for discussing the new isotopic data in a well-constrained framework. Because of disparate results across the various

proxies, there remain uncertainties about the timing and factors that induced major environmental changes and how they

affected the evolution of fauna, including humans in the Plio-Pleistocene in the Shungura Fm and more widely in eastern Africa, (e.g., Trauth et al., 2021 for a recent review).

By determining to what extent the $\delta^{18}O_p$ of crocodilians would allow tracking complex environmental changes in this region, this paper discusses (1) the diversity of aquatic environments occupied by crocodilians by interpreting the range of $\delta^{18}O_p$
values, and (2) the variation of the $\delta^{18}O_p$ values as an indicator of local evaporation and $\delta^{18}O$ values of meteoric waters. Building up on that discussion, guidelines are provided for further applications to other paleontological and archaeological sites.

## 2 Material and method

### 2.1 Study area and main information on the aquatic environments

The Omo Group, including the Shungura Fm, became a major reference for the Plio-Pleistocene biotic and environmental
evolution thanks to the early work of the International Omo Research Expedition (IORE) from 1967 to 1976 (e.g., Howell, 1968; Coppens & Howell, 1976; Heinzelin, 1983). The present study takes place within the framework of renewed fieldwork in the Shungura Fm and Usno Fm enabling new paleontological, archaeological, geological, and paleoenvironmental analyses developed by the Omo Group Research Expedition (OGRE) on both IORE and new data. This research program investigates biological evolution and its interactions with environmental factors at the local scale of the Lower Omo Valley (Boisserie et
al., 2008). One approach favoured by the OGRE is to exploit paleoenvironmental proxies directly linked to the continental paleobiological record (e.g., Souron et al., 2012; the present study).

The Shungura Fm is exposed in the Lower Omo Valley (Fig. 2). With 800 m of alternating volcanic tuffs and fluvial, deltaic and lacustrine fossil-bearing sediments (more than 57,000 specimens to date), this continental sedimentary succession is one

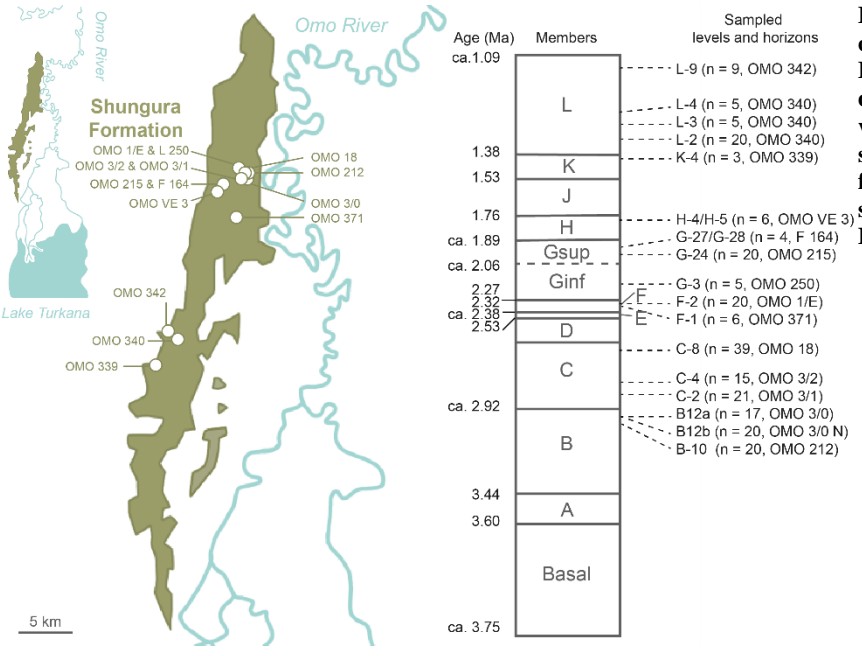

**Figure 2. Geographic location and chronostratigraphic column of the Shungura Formation, Lower Omo Valley, Ethiopia and distribution in the map and the log of horizons where crocodilian teeth were sampled for this study. Geographic location of geochemical data from East-African GNIP stations and water samples of Ethiopian rivers and lakes (map ESRI 2022).**

of the most extensive and best-dated, documenting the evolution of ecosystems of the Lower Omo between ca. 3.75 Ma and ca. 1.09 Ma (Fig. 2). The stratigraphic division into members and sub-members of the Shungura Formation defined by Heinzelin (1983) is used in this paper, while keeping in mind that it has its limits at the scale of the formation. A revised age model including the latest absolute ages is provided in Table S1. For the last 5 million years, the Omo River, which also feeds Lake Turkana, mainly received sediments and water from the Ethiopian Highlands. The Shungura Fm recorded hydrological change over time: although it was dominated by a river system most of the time, the level of the paleolake occupying the Turkana Depression rose until covering the site in the middle of Member G (2.06-1.95 Ma, Lorenyang highstand) and again at the top of Member L (1.19-1.09 Ma) (Haesaerts et al., 1983; Nutz et al., 2020). New fieldwork observation by the OGRE suggests that the Shungura Fm was under lacustrine influence from the Basal Member to unit B-8, maybe until B-10 (3.75-2.98 Ma, Lokochot highstand). Currently, the course of the Omo River is more incised and constrained due to the rift narrowing that occurred after 1.2 Ma (Nutz et al., 2022). Before, the Omo River migrated onto the area occupied at present-day by the Shungura Fm.

This area underwent a general trend towards aridification in the Plio-Pleistocene, with the Lower Omo Valley becoming increasingly dominated by grasslands (e.g., Levin et al., 2011; Negash et al., 2015, 2020; Blondel et al., 2018, 2022). This trend was nevertheless locally buffered in the Lower Omo Valley, which maintained a more wooded environment than along the lake shores (Nachukui Fm and Koobi Fora Fm), at least until the end of the Lorenyang highstand (Levin et al., 2011). This was punctuated by periods of increased humidity and precipitation in the Ethiopian Highlands, which temporarily increased the Omo woody cover (Nutz et al., 2020). This may have made the Omo Valley a refuge for forest and woodland flora and fauna during this period of high climate variability (e.g., Bobe, 2006).

### 2.2 Plio-Pleistocene eastern African crocodilian diversity

The Shungura Fm yielded two distinct crocodilian teeth morphologies: elongated and anteroposteriorly curved teeth (hereafter referred to as pointed) and shorter and stouter, conical teeth (hereafter referred to as stout) (Fig. 3). This indicates the persistent presence of several crocodilian taxa over time, despite the local environmental changes. Three main genera of crocodilians have been identified in the Plio-Pleistocene outcrops of the Turkana Depression: the longirostrine and piscivorous *Euthecodon* and *Mecistops*, and the generalist to brevirostrine *Crocodylus* (*C. niloticus* preys on all kind of prey and *C. thorbjarnarsoni* is the largest known species in this genus and probably fed on large mammals) (Arambourg, 1967; Tchernov 1976, 1986; Storrs, 2003; Brochu et al., 2010; Brochu and Storrs, 2012; Brochu, 2020). The pointed crocodilian teeth correspond to a piscivorous crocodilian such as *Euthecodon* and *Mecistops*, whereas the stout teeth could belong to *Crocodylus,* and possibly *Mecistops* too (molariform teeth at the back of the tooth rows). The different diets reflected by these two dental morphologies could induce different ecologies and therefore lead to a different oxygen isotopic signal as well, which was also tested here.

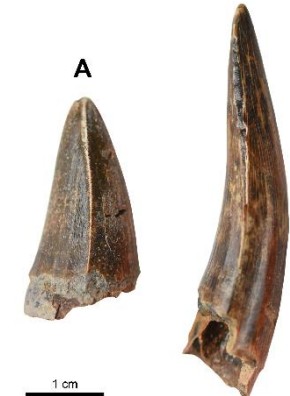

**Figure 3. Two morphotypes of crocodilian teeth are encountered in the Shungura formation outcrops: A, stout (B-12-r-1); B, pointed (B-12b-p-2).**

## 2. 3 Sample collection

Two hundred thirty-eight (238) crocodilian teeth were collected from 16 units of the Shungura Fm aged between ca. 2.97 Ma and ca. 1.14 Ma (Tables 1 and S2, Fig. 2). The specimens were purposely collected for destructive analysis during the OGRE field missions in 2013 and 2016. To determine the minimum number of teeth required for a reliable representation of isotopic range of values, an approach based on data from level C-8 was employed chosen for its larger sample and its large range of the values. By applying the slope break criterion, a threshold of 6 teeth was established as optimal to stabilize the range measurement. However, in 5 levels, fewer teeth were available, and it is acknowledged that this requires increased caution in interpreting these results. All teeth with a minimum size of 1 cm in height were exhaustively collected (stout and pointed) from a large part of the Shungura series, each sample corresponding to a single locality (or part of a locality). This sampling strategy allows for exploring the signal yielded by crocodilian teeth in units documenting major environmental transitions. Units K-4 and L-3 did not yield stout teeth, and no pointed teeth were present in unit L-9. The analyses presented were preferentially conducted on pointed teeth, to build a comparable dataset along the series, as they are most likely to be from piscivorous crocodilians and are particularly linked to aquatic environments. Teeth with bright enamel and without white alteration marks were selected for isotopic analyses. B-12a and B-12b are two spots in the same locality in lateral continuity, OMO 3/0 (ca. 2.94 Ma), separated by about 200 m, with B-12a being siltier and more stratified and B-12b coarser and with less structure. Some units are sometimes difficult to distinguish, and the elements collected on a surface can thus group two units (Tables 2 and S2, Fig. 2). On surface collections of this type, a little temporal mixing is therefore possible. The average height of 'stout' teeth is approximately 1.8 cm, while 'pointed' teeth have an average height of around 3.0 cm. A crocodile is typically considered large when it reaches a mass of over 100 kg and an estimated length of at least 2.8 meters (Smith, 1979). Using approximate calculations based on photographs of skulls and live specimens, the minimum size of the crocodilians represented in the sample is approximately two meters, but the majority of them likely exceed 2.8 meters in length, classifying them as large individuals.

## 2. 4 Sample preparation and analysis

For each tooth, enamel was sampled from base to apex of the crown to average potential seasonal variation in $\delta^{18}O_p$ using a micro-milling Dremel device and was then crushed in an agate mortar and pestle.

To isolate the phosphate group of enamel apatite (O'Neil et al., 1994; Wenzel et al., 2000), dissolved apatite was converted in $Ag_3PO_4$ following a method adapted from Joachimski et al. (2009). About one mg of enamel powder was weighed into small polyethylene beakers and dissolved overnight in 33 µl 2M $HNO_3$. The solution was neutralized with 33 µl 2M KOH, and $Ca^{2+}$ precipitated as $CaF_2$ by adding 33 µl of 2M HF. After centrifugation, the supernatant was transferred to a clean beaker and 0.5 ml of a silver amine solution was added in order to precipitate $Ag_3PO_4$ at 60°C for 8 hours. Silver phosphate crystals were rinsed several times with distilled water and dried at 40°C for 30-60 minutes, and ground into powder in an agate mortar and pestle.

Between 0.2 mg and 0.3 mg of $Ag_3PO_4$ were weighted into silver foil and analysed for part of the sample set at the GeoZentrum Nordbayern of the Friedrich-Alexander University Erlangen-Nürnberg using a TC-EA (high-temperature elemental analyser) coupled online to a Thermo Fisher Delta V IRMS and for the other part at the University of Burgundy (Dijon, France) using a High-Temperature Pyrolysis Analyzer (Elementar Pyrocube) connected online to an Elementar Isoprime mass spectrometer. Samples were all combusted at 1450 °C and the generated CO gas was transferred in a helium stream to the mass spectrometer. Samples as well as standards measured over the course of analyses were generally measured in triplicate. All values are reported in ‰ relative to V-SMOW. The average oxygen isotopic composition of NBS 120c standard was measured as 21.7 ‰ VSMOW. Accuracy was monitored by replicate analyses of NBS120c. Reproducibility was calculated based on triplicate sample and standard analyses and was generally < ±0.3 ‰ (based on 1SD statistics).

Oxygen isotope analyses have additionally been performed on the carbonate group of some powdered enamel samples ($\delta^{18}O_c$) to check for a possible effect of diagenesis on the sample isotopic composition. To this end, 0.10-0.13 mg of enamel powder were loaded in glass vials and reacted with a 102% $H_3PO_4$ solution at 70 °C for 20 min in a ThermoScientific™ Kiel VI carbonate preparation device coupled with a ThermoScientific™ Delta V Plus™ IRMS at the Biogéosciences Laboratory of the University of Burgundy. $\delta^{18}O_c$ data were reported in ‰ relative to V-PDB (Vienna Pee Dee Belemnite) using certified international standard NBS19 ($\delta^{18}O = -2.20$ ‰). External reproducibility was assessed by replicate analyses of NBS19 over the course of the analyses and is better than ±0.07 ‰ (2σ). Calculation of $\delta^{18}O_w$ values from $\delta^{18}O_p$ of Shungura fossil crocodilian enamel is based on the fractionation equation of Amiot et al. (2007).

## 2. 5 Statistical analyses

Means are presented with ±1- standard deviation. Significance is evaluated at α = 0.05. Analyses were performed in R Version 4.2.2 (R Core Team, 2022). The $\delta^{18}O_p$ difference between stratigraphical units was investigated with a two-way ANOVA and Pairwise Wilcoxon Rank Sum Tests with Holm correction (non-parametric post-hoc tests). The trend over time was identified using least-squares linear regression and reported as adjusted $R^2$.

## 3 Results

The isotopic composition of crocodilian enamel phosphate and the resulting estimated $\delta^{18}O_w$ are summarised in Table 1 (full dataset in Table S2) and displayed in Figs. 4 and 8. $\delta^{18}O_p$ values range between 15.1 ‰ and 22.9 ‰, while $\delta^{18}O_c$ values range between −8.2 ‰ and 1.2 ‰. The $\delta^{18}O_p$ is not statistically different between round and pointed teeth (ANOVA: F = 1.136, df= 8, p = 0.342) (Fig. 4).

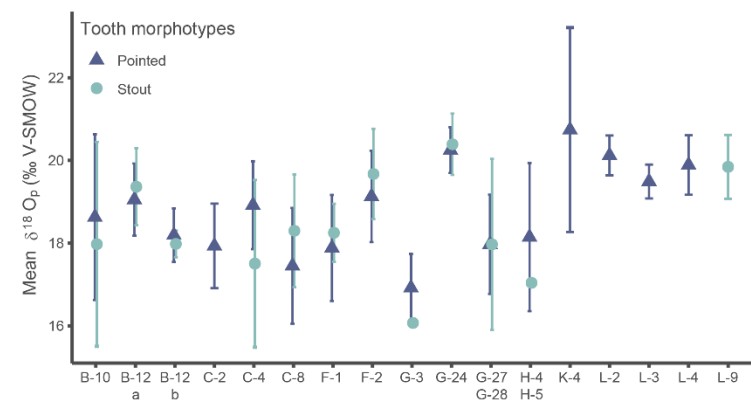

**Figure 4. Oxygen isotopic composition (mean±1sd) of the crocodilian teeth of the Shungura Formation, relative to the tooth morphotypes.**

From B-10 to C-8, F-2, G-27/G-28, H-4/H-5, and K-4, the range of $\delta^{18}O_p$ values is greater than 2.3 ‰; while in the other units F-1, G-3 to G-24, and L-2 to L-9, the range of $\delta^{18}O_p$ values does not or barely exceeds this variation. Minimal $\delta^{18}O_p$ values increase from 15.3 ‰ in B-10 (2.97 Ma) to 18.6 ‰ in L-9 (ca. 1.14 Ma) ($R^2$=0.49), corresponding to an increase in $\delta^{18}O_w$ values from −6.6 ‰ to −3.5 ‰. However, this trend is nuanced when studying the isotopic signal between the stratigraphic units. The $\delta^{18}O_p$ values are particularly high in units F-2, G-24, L-2, and L-9, while units C-2 and C-8 yielded low $\delta^{18}O_p$ values (Table S3). The $\delta^{18}O_p$ values are statistically homogenous in the two sub-localities of unit B-12 (spatially close but with slightly different sedimentary facies, p = 0.12). After unit H-4/H-5 (1.84 Ma), the $\delta^{18}O_p$ values strongly increase and remain high.

**Table 1. Mean, SD, minimal and maximal values of $\delta^{18}O_p$ values for each sampled locality of the Shungura Formation. Abbreviations: n number of analysed teeth.**

| Unit | Locality | Age (Ma) | n | $\delta^{18}O_p$ of crocodilian teeth (‰ V-SMOW) | | | | | Estimated $\delta^{18}O_w$ (‰ V-SMOW) | |
| --- | --- | --- | --- | --- | --- | --- | --- | --- | --- | --- |
| | | | | Mean | SD | Min | Max | Range | Min | Max |
| L-9 | OMO 342 | ca. 1.14 | 9 | 19.84 | 0.77 | 18.55 | 20.95 | 2.4 | −3.92 | −3.44 |
| L-4 | OMO 340 | ca. 1.24 | 5 | 19.89 | 0.72 | 19.28 | 21.00 | 1.72 | −3.32 | −1.91 |
| L-3 | OMO 340 | ca. 1.30 | 5 | 19.49 | 0.41 | 19.07 | 20.13 | 1.06 | −3.49 | −2.62 |
| L-2 | OMO 340 | ca. 1.34 | 20 | 20.12 | 0.48 | 18.99 | 20.78 | 1.79 | −3.56 | −2.09 |
| K-4 | OMO 339 | ca. 1.39 | 3 | 20.74 | 2.47 | 18.06 | 22.93 | 4.87 | −4.32 | −0.34 |
| H-4/H-5 | OMO VE 3 | 1.84 | 6 | 17.96 | 1.66 | 16.29 | 20.99 | 4.7 | −5.77 | −1.92 |
| G-27/G-28 | F 164 | ca. 1.95 | 4 | 17.97 | 1.20 | 16.01 | 20.48 | 4.47 | −6.00 | −2.34 |
| G-24 | OMO 215 | ca. 1.98 | 20 | 20.28 | 0.58 | 19.40 | 21.28 | 1.88 | −3.22 | −1.68 |
| G-3 | L 250 | 2.19 | 5 | 16.75 | 0.80 | 16.07 | 18.04 | 1.97 | −5.95 | −4.34 |
| F-2 | OMO 1/E | ca. 2.32 | 20 | 19.45 | 1.10 | 16.57 | 20.87 | 4.3 | −5.54 | −2.01 |
| F-1 | OMO 371 | ca. 2.32 | 6 | 17.88 | 1.28 | 16.46 | 18.97 | 2.51 | −5.63 | −3.57 |
| C-8 | OMO 18 | ca. 2.57 | 39 | 17.86 | 1.43 | 15.11 | 21.08 | 5.97 | −6.74 | −1.84 |
| C-4 | OMO 3/2 | ca. 2.76 | 15 | 18.57 | 1.43 | 15.87 | 20.26 | 4.39 | −6.12 | −2.52 |
| C-2 | OMO 3/1 | ca. 2.86 | 21 | 17.96 | 1.00 | 15.84 | 19.70 | 3.86 | −6.14 | −2.98 |
| B-12a | OMO 3/0 | ca. 2.94 | 17 | 19.11 | 0.86 | 17.92 | 20.89 | 2.97 | −4.45 | −2.00 |
| B-12b | OMO 3/0 N | ca. 2.94 | 20 | 18.14 | 0.58 | 16.85 | 18.96 | 2.11 | −5.31 | −3.59 |
| B-10 | OMO 212 | 2.97 | 20 | 18.53 | 2.02 | 15.32 | 22.22 | 6.9 | −6.56 | −0.91 |

## 4 Discussion

### 4.1 Preservation of the original oxygen isotopic composition

Tooth enamel has densely packed large apatite crystallites, which should best retain the primary $\delta^{18}O_p$ (Kolodny et al., 1983; Shemesh et al., 1983; Pucéat et al., 2004). However, alterations due to secondary apatite precipitation and diagenesis can still occur and the preservation of the original isotopic signal must be ensured before proceeding to paleoenvironmental interpretations (Blake et al., 1997; Zazzo et al., 2004). Although no dual $\delta^{18}O$ data exist for modern crocodilians or other sauropsids, which could differ from mammals, the data in this article are compared to those obtained from well-preserved modern mammals and Mesozoic vertebrates from Morocco (Iacumin et al., 1996; Amiot et al., 2010b). The fossil data from Amiot et al. (2010b) exhibit a lack of homogenization in $\delta^{18}O$ values and follow an expected linear relationship for at least

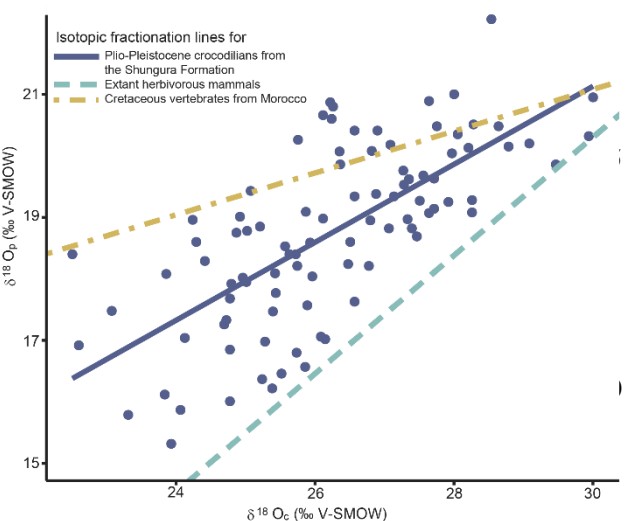

**Figure 5. Oxygen isotope compositions of enamel phosphate ($\delta^{18}O_p$) from the studied crocodilians from the Shungura Formation plotted against their corresponding oxygen isotope composition of carbonate ($\delta^{18}O_c$), along with the phosphate–carbonate isotopic fractionation line established for Cretaceous vertebrates from Morocco (Amiot et al., 2010b) and extant herbivorous mammals (Iacumin et al., 1996).**

partially preserved isotopic compositions. Applying the same approach to the crocodilians of the Shungura Formation demonstrates at least partial preservation of the original signal by exhibiting a linear relationship. While achieving an exact match can be challenging due to various factors, including differences in clades' evolutionary history, the new dataset falls within the anticipated range for values considered "at least partially preserved" (adjusted $R^2 = 0.50$; Spearman's correlation coefficient = 0.74) (Fig. 5), and thus supports reasonable interpretability. In addition, the lack of homogeneity (< 1 ‰ range) of $\delta^{18}O_p$ values within each unit further supports this view, reflecting the diversity of the life history of individuals and the good match of values to other geochemical studies dealing with modern mammals and ancient crocodilians (Iacumin et al., 1996; Amiot et al., 2010b). The $\delta^{18}O_p$ values of the teeth analysed in this work can be interpreted as a function of the evolution of the local aquatic environments and regional climate during the Plio-Pleistocene in the Turkana Depression.

## 4.2 Construction of the interpretation model of the $\delta^{18}O_p$ of a fossil crocodilian enamel in the Shungura Fm

### 4.2.1 How representative is the $\delta^{18}O_p$ of a fossil crocodilian tooth of the water body it used to live in?

To use the $\delta^{18}O_p$ of crocodilian enamel as a paleoenvironmental proxy, it is first necessary to understand how the isotopic signal is constructed. The $\delta^{18}O_p$ of crocodilian enamel depends on the body temperature in the jaw where the enamel mineralizes, which controls the amplitude of the isotopic fractionation between phosphate and body water, and on the isotopic composition of the body water, which is dependent on the balance between oxygen inputs and output fluxes (Kohn, 1996; Bryant and Froelich, 1995; Amiot et al., 2007) (Fig. 6).

Crocodilians have various sources of oxygen, including drinking water, water from food, oxygen from dry food, and inhaled $O_2$. Oxygen outputs include liquid water losses (urine) and vapour water losses (cutaneous, oral, and nasal) as well as exhaled $CO_2$ (Bryant and Froelich, 1995; Kohn, 1996). Rough estimates of the relative contribution of these oxygen fluxes can be made based on the existing literature on crocodilian physiology and ecology (Cott, 1960; Cloudsley-Thompson, 1968; Altman & Dittmer, 1972; Grigg, 1978; Glass & Johansen, 1979; Seymour et al., 1985, 2013; Davis et al., 1980; Mazzotti & Duson, 1989; Christian et al., 1996). Based on these studies, at 25 °C, 85 % of oxygen input comes from environmental water and most of

the water output is vapour water losses (up to 75 %) (Fig. 6). Hence, the crocodilian body water $\delta^{18}O$ is mainly determined by their drinking water, corresponding to the water in which crocodilians live ($\delta^{18}O_w$), and by body water evaporation (Fig. 6).

Crocodilians are ectothermic vertebrates, meaning their body temperature fluctuates with environmental conditions, which has an impact on the amplitude of the oxygen isotopic fractionation between body water and enamel apatite. However, they can somewhat regulate their body temperature by land-water movements for heat-seeking or -avoidance and physiological adjustments (Smith, 1979; Lang, 1987a, 1987b). The activity range for many extant species is 25-35 °C, but they tend to prefer a thermal gradient near the upper limit (35-37 °C) in tropical regions (Lang, 1987b; Markwick, 1998). Existing studies suggest

that crocodilians have a high degree of thermoregulation, and their body temperature is relatively independent of water temperature, especially for large individuals ($\geq 100\,kg$) (Smith, 1979; Lang, 1987b; Grigg et al., 1998; Seebacher et al., 1999). This is due to the convective transfer of heat decreasing and the metabolic production of heat increasing with body size. In this study, most of the sampled material consists of large fossil teeth, allowing to assume that they may have belonged to large individuals with relatively stable and high body temperatures. Given the thermoregulation abilities of crocodilians, depending

on the individuals and their behaviour and environment, the body temperature can be slightly different between individuals occupying the same water body, and possibly greater between two populations and species, occupying different aquatic environments (Amiot et al., 2007). The body temperature of crocodilians varies mainly in a range of 10 °C which approximately corresponds to a difference of ~2.3 ‰ in $\delta^{18}O_p$ (Longinelli and Nuti, 1973; Kolodny et al., 1983; Pucéat et al., 2010) (Fig. 6). Amiot et al. (2007) did not identify any interspecific variation of $\delta^{18}O_p$ in captive crocodilians reared in the

same water, but it may be due to the artificial environment they were in, which may not have allowed them to display the same diversity of behaviour as in the wild.

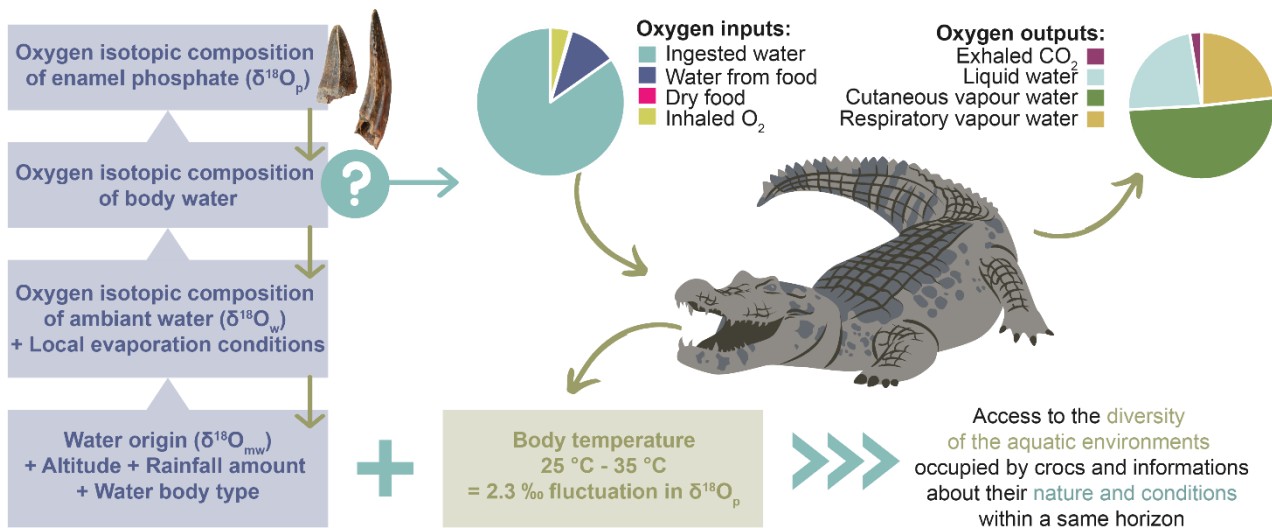

**Figure 6. Summary of the extrinsic (water origin, altitude, rainfall amount, evaporation) and intrinsic (oxygen outputs, body temperature) factors influencing the oxygen isotopic composition of the crocodilian enamel, including the main oxygen fluxes controlling the oxygen isotope composition of crocodilian body water.**

Dental development can also influence the construction of the isotopic signal, in particular the duration of enamel mineralization and time of year can influence the duration of recording of seasonal isotopic fluctuations in water. According to previous studies, tooth replacement in alligators, occurs approximately once a year, with a range of eight to sixteen months, depending on the tooth's position in the dental row (Edmund, 1962). Other studies, such as those by Erickson (1992, 1996), based on dentin increment counting, estimate tooth formation time in both modern and fossil crocodilians to be between 83 and 285 days and seem to increase with age and body size. Therefore, it is difficult to definitively state whether tooth formation occurs during a particular season, and it is important to note that crocodilians may estivate or hibernate, which can affect somatic and maybe dental growth (Chabreck and Joanen, 1979; Hutton, 1987; Taplin, 1988).

Crocodilians are relatively sedentary, they only migrate over short distances seasonally or to mate (a few tens of kilometres) (Lang, 1987a; Tucker et al., 1997; Swanepoel, 1999; Campbell et al., 2010; Beauchamp et al., 2018). Their fast-growing teeth thus record the oxygen isotopic composition of the water body or watershed they occupied. Despite the significant Plio-Pleistocene hydrological changes, several crocodilian species have persisted for millions of years in the Shungura Fm, likely due to their ability to adapt to a wide range of aquatic habitats. These include rivers, swamps, lakes, estuaries, and freshwater and saline waters (e.g., Isberg et al., 2019). The diversity of water bodies occupied by past crocodilians thus likely reflects a significant part of the types of aquatic environments available at a local scale (Figs. 1 and 6).

### 4.2.2 Determinants of $\delta^{18}O$ of Ethiopian waters

Rainfall amount, topography, distance from coastlines, temperature and moisture source are usually invoked to explain the isotopic composition of meteoric waters ($\delta^{18}O_{mw}$) in intertropical regions (Dansgaard, 1964; Rozanski et al., 1996; Levin et al., 2009). Moisture supply in the Ethiopian Highlands is determined by the seasonal migration of the Intertropical Convergence Zone (ITCZ) and the Congo Air Boundary (CAB) and comes from the Atlantic and Indian Oceans, recycled continental moisture from the Congo Basin and possibly from southern lakes (Levin et al., 2009; Kebede and Travi, 2012). Table 2 shows that the oxygen isotopic composition of rainfalls in eastern Africa at similar altitudes is not correlated to the amount of precipitation. This indicates that other factors, such as the source of the moisture, are dominant.

Currently, the Turkana Depression is a region with a hot and semi-arid climate, with less than 500 mm of annual precipitation and an annual average temperature of 29-30 °C (Merkel, 2023), a climate that does not explain the relatively large water availability in the basin. Lake Turkana is mainly fed by the Omo River, which originates in the Ethiopian Highlands, southwest of Addis Ababa (Schuster et al. 2022). The $\delta^{18}O_w$ of the Omo River is influenced by the climate of this region, where moisture is brought at 75 % by the Western African Monsoon, coming from the Atlantic Ocean and crossing the Congo basin (June-September, $\delta^{18}O_{mw}$ of −2.59 to −1.99 ‰ at Entebbe, west to Addis Ababa), and the rest by the Indian Summer Monsoon from the equatorial Indian Ocean ($\delta^{18}O_{mw}$ of −4.65 to −3.81 ‰ at Muguga, east to Addis Ababa) (Camberlin, 1997; Kebede and Travi, 2012). In regions with high levels of evapotranspiration, the $\delta^{18}O_{mw}$ values are generally similar or greater than the $\delta^{18}O$ values of ocean source water due to the wholesale return of transpired water consumed by plants to the atmosphere (Levin et al., 2009). Therefore, the water vapour in the Congo basin results in rainfall with $\delta^{18}O_{mw}$ equal to or greater than that of the

Atlantic Ocean. This [18]O-enriched rainfall is the main contributing factor to the higher $\delta^{18}O_{mw}$ values recorded in Addis Ababa compared to other parts of eastern Africa (Table 1, Rozanski et al., 1996; Levin et al., 2009; Kebede and Travi, 2012). The Ethiopian and eastern-African Dome form channels or obstacles for these low-level air streams, preventing the Indian Summer Monsoon to get further west in the Ethiopian Highlands and the Atlantic and Congo moisture to get further north and east (Levin et al., 2009). The elevation of the Ethiopian Dome up to 4500 m has mainly taken place well before the Plio-Pleistocene, so a change in topography seems unlikely to result in significant variation in $\delta^{18}O_{mw}$ in the context of the Shungura series (Abbate et al., 2015). Moreover, Passey et al. (2010) concluded that air temperature has not changed or only slightly decreased in the Turkana Depression over the past 3 million years, and thus cannot explain the $\delta^{18}O_{mw}$ variation observed in this region.

**Table 2. Present-day geographic, meteorological, and geochemical data recorded from eastern African GNIP stations and surface water samples of Ethiopian rivers and lakes, placed on the map in Fig. 1A. Additional data from the Goma station is provided for the Discussion section.**

| Locality | Country | Altitude (m) | Mean annual precipitation (mm) | Mean annual $\delta^{18}O$ (‰, V-SMOW) | References |
|---|---|---|---|---|---|
| *Surface waters* | | | | | |
| Omo Delta | Ethiopia | 395 | 226 | −1.2 to −0.1 | Vonhof et al., 2013 |
| Lake Turkana | Kenya | 360 | 437 | 4.44 to 7.21 | |
| Awash River | Ethiopia | 1590 | 371 | −2.41 | Hailemichael et al., 2002 |
| Lake Koka | Ethiopia | 1590 | 371 | 1.55 | |
| *Meteoric waters from eastern Africa* | | | | | |
| Addis Ababa | Ethiopia | 2355 | 1179 | −1.31 | |
| Dar Es Salaam | Tanzania | 55 | 1123 | −2.83 | |
| Entebbe | Uganda | 1140 | 1628 | −2.78 | |
| Hagere Selam | Ethiopia | 2625 | 1138* | −3.34 to −0.67 | IAEA, 2023 |
| Kericho | Kenya | 2144 | 1958* | −2.77 | except *Merkel, 2023 |
| Moyale | Ethiopia | 1090 | 619* | −4.04 to −3.82 | |
| Muguga | Kenya | 2102 | 674* | −6.18 to −1.09 | |
| Rukungiri | Uganda | 1640 | 1509* | −4.78 to −0.11 | |
| *Meteoric waters from central Africa* | | | | | |
| Goma (Biraro) | Democratic Republic of the Congo | 1540 | 1916 | −3.55 | IAEA, 2023 |

Today, the average $\delta^{18}O_w$ value at the Omo River Delta is between −1.2 ‰ and −0.1 ‰, while the value at Lake Turkana (endorheic) is between 4.4 ‰ and 7.2 ‰, with an [18]O-enrichment between 4.3 ‰ and 8.4 ‰ (Table 1) (Kebede and Travi, 2012; Vonhof et al., 2013) (Table 1). Lake Koka is an artificial and exoreic lake ($\delta^{18}O_w$ = 1.55 ‰) with a minor [18]O-enrichment of 4.0 ‰ compared to its tributary, the Awash River ($\delta^{18}O_w$ = −2.4 ‰) (Table 1) (Hailemichael et al., 2002). The surface area, evaporation and drainage can be determining factors for the isotopic composition of lakes (Kebede et al., 2009; Kebede and Travi, 2012). From the data of Lake Koka and the Awash River, one could expect an exoreic lake fed by the Omo River to have a $\delta^{18}O_w$ in the order of 2 ‰ to 4 ‰. These values will be compared to the Plio-Pleistocene $\delta^{18}O_w$ values estimated from

the $\delta^{18}O_p$ of Shungura crocodilian enamel. Calculation of $\delta^{18}O_w$ values from $\delta^{18}O_p$ of Shungura fossil crocodilian enamel is
315 based on the fractionation equation of Amiot et al. (2007). Note that this fractionation equation mainly relies on isotopic data
from extant modern captive crocodilians, which form their teeth in humid greenhouse contexts, potentially overestimating the
inferred values. In their natural environment, large cutaneous and respiratory evaporation may lead to an $^{18}O$ enrichment of
crocodilians' body water relative to their environmental water, which is not considered in the equation (Amiot et al., 2007).

### 4.2.3 Interpretative model of the $\delta^{18}O_p$ of crocodilian enamel

The interpretative model proposed in this paper is adapted to the context of the Shungura Fm (Fig. 7). The first part of the
model is based on the idea that the teeth discovered in the Shungura Formation may have originated from two scenarios: (1)
some teeth belonged to individuals that developed their teeth within the local waterbody, reflecting the depositional
environment of the Shungura Formation, or (2) other teeth might have come from individuals growing their teeth in another
nearby waterbody, and these teeth were later shed when the crocodilian came to the waterbody recorded in the Shungura Fm.
Considering a ~2.3 ‰ range in $\delta^{18}O_p$ will allow to take precaution for possible body temperature differences between
individuals (behaviour, size, age), populations, and species. Once this variation in $\delta^{18}O_p$ due to the fluctuating body temperature
of crocodilians is excluded, the remaining range of $\delta^{18}O_p$ values within the same unit should reflect the local variety of water
body types (with different $\delta^{18}O_w$) occupied by these animals. In other words, only one type of aquatic environment was

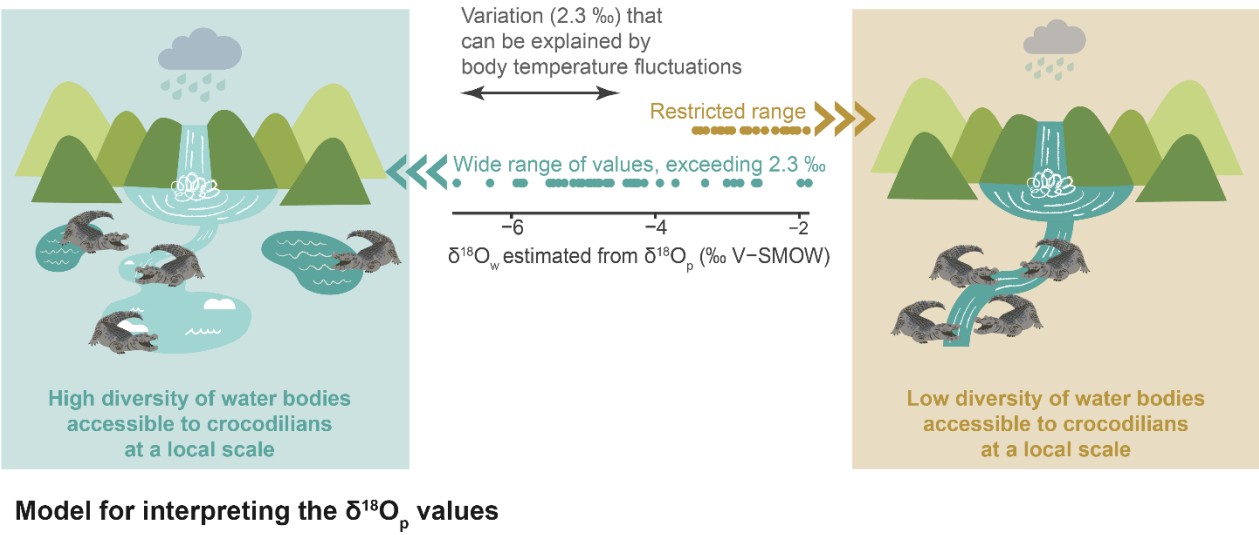

**Figure 7. Interpretative model of the $\delta^{18}O_w$ estimated from $\delta^{18}O_p$ of crocodilian teeth in the context of the Shungura Fm.**

available for crocodilians in a unit where the data distribution does not exceed ~2.3 ‰. On the contrary, if the distribution exceeds ~2.3 ‰, then different aquatic environments were occupied by these animals nearby, characterised by different evaporation conditions and/or different water sources. This part of the model opens a window into the hydrological dynamics of the basin. It is noteworthy that this variability in the paleohydrology, revealed by a single stratigraphic unit-based study, corresponding to or within a given locality, could not have been highlighted by a study based on mean values performed at the stratigraphic member-based scale of the Shungura Formation. As mentioned in section 4.2.1, the season and duration of tooth mineralization cannot be definitively determined but might affect the interpretation of the resulting isotopic signal. Hence, the diversity of freshwater environment observed with the range of $\delta^{18}O_w$ could be on an annual or seasonal scale within each unit. Moreover, the interannual variability of the isotopic signal in the Turkana Depression can be influenced by different types of aquatic environments and water mixing, as evidenced by mollusc shells in the Omo delta (range of 6 ‰) Omo River (range of 2.5 ‰); Turkana Lake (negligible fluctuations) (Vonhof et al., 2013).

Concerning the second part of the model, interpreting past $\delta^{18}O_w$ in the Omo Valley context should mainly consider the variability of the evaporative state of water (aridity, drainage, and depth of water bodies) and rainfall (especially the water supply on the Ethiopian Dome), low-level air streams and the migration of the convergence zone. In other words, decreasing $\delta^{18}O_w$ values might be mainly linked to an increased rainfall supply on the Ethiopian Dome, a northward penetration of the Indian Summer Monsoon (consistent with low relief and a decrease in its intensity), and/or a more open/deep water body (Otero et al., 2011; Kebede and Travi, 2012). On the contrary, increasing $\delta^{18}O_w$ values are expected to result from drier regional conditions, an increase in the intensity of the Indian Summer Monsoon and a relatively greater contribution of the recycled moisture from the Congo Basin to the water supply, and/or to a close and/or shallow water system (Otero et al., 2011; Kebbede and Travi, 2012).

## 4.3 Interpretation of the $\delta^{18}O_p$ of fossil crocodilian enamel

### 4.3.1 Restriction of local diversity of aquatic environments

The large range of $\delta^{18}O_p$ values in units B-10 to C-8 indicates that crocodilians occupied different water body types. Low $\delta^{18}O_p$ values suggest water bodies with low evaporation state (probably linked to short water residence time, through flowing, exoreic lake basin, river setting) or an $^{18}O$-depleted water source (e.g., unevaporated meteoric water from rivers or underground water); while the high values reflect shallow and/or isolated water bodies subject to strong evaporation (e.g., enclosed endoreic lake basin, pond, shallow small lake, swamp) or an $^{18}O$-enriched water source. If the Shungura Fm was under lacustrine influence in unit B-10, it could explain the great diversity of water bodies present in the area (Fig. 8). In Member C, the environment is predominantly fluvial, but the present isotopic data suggest that it was probably connected to other nearby water bodies making them accessible to local crocodilians. These $\delta^{18}O_p$ values do not indicate aridification affecting freshwater environments between 2.8 Ma and 2.5 Ma (Members B and C). This contrasts with changes observed in the aquatic and terrestrial ecosystems of the eastern African Rift: such as lake level drop, landscape opening, and aridification (e.g., Maslin and Trauth, 2009; Trauth

et al., 2009, 2021; Levin et al., 2011; Negash et al., 2015, 2020; Blondel et al., 2018, 2022; Nutz et al., 2020) (Fig. 8). Interestingly, the $\delta^{13}C$ of soil carbonates increases, thus indicating aridification in the Shungura Fm but not in the Nachukui Fm in the West Turkana area (Levin et al., 2011; Levin, 2013; Nutz et al., 2020) (Fig. 8). Nutz et al. explain (2020) that the $\delta^{13}C$ record in the Shungura Fm reflects climatic conditions upstream on the Ethiopian Dome, and that of Nachukui Fm reflects more local and regional climatic conditions. The $\delta^{13}C$ of the paleosols would therefore indicate a less significant supply of water from the Ethiopian Dome towards the Lower Omo Valley. On the other hand, the hydrological conditions remain stable in the western part of the Turkana Depression between 2.8 Ma and 2.5 Ma, thus possibly preserving a certain diversity of aquatic environments in the surroundings. Instead of increasing and showing a restricted range of $\delta^{18}O_p$ values, as expected for an aridification context, the distribution of $\delta^{18}O_p$ values in Members B to C remains relatively stable, suggesting that crocodilians had access to the same range of water bodies, which remained nearby or interconnected. It implies that the $\delta^{18}O$ of the water sources remained relatively the same; or less likely, if evaporation increased, the $\delta^{18}O$ of the source decreased in balance (resulting then from a greater contribution of the Indian Summer Monsoon) (Kebede and Travi, 2012) (Fig. 8).

The $\delta^{18}O$ values are more homogeneous in unit F-1, indicating that crocodilians probably had access to only one type of water body at the local scale (Fig. 8). The minimal values are higher than in the older units, while the maximal values are lower, indicating that if the most evaporated water bodies present in Members B and C still existed, they were no longer accessible to crocodilians. In unit F-2, the values are mostly higher than those of F-1 except for two teeth. This range would reflect at least two distinct water body types (Fig. 8). This is coherent with the formation of this unit during a period of lake transgression and increased water supply over the Ethiopian Dome (Levin et al., 2011; Nutz et al., 2020) (Fig. 8).

By contrast, in unit G-3 (with few teeth analysed), crocodilian teeth yielded a restricted range of low $\delta^{18}O_p$ values, reflecting a single accessible environment, probably the river according to sedimentological studies and to the increasingly humid conditions (Haesaerts et al., 1983; Levin et al., 2011; Nutz et al., 2020) (Fig. 8).

During the Lorenyang lake's highstand, the lake reached its maximum extent between approximately 2.10 Ma and 1.90 Ma, coinciding with the presence of lacustrine facies in the Shungura Formation. According to Nutz et al. (2020), the regression phase commenced around 2 Ma (with G-24 dated at approximately 1.98 Ma), marked by a shoreline migration towards the basin and the gradual reappearance of deltaic facies within the G-sup Member. This suggests a transition towards more segmented aquatic environments, including distributary channels, abandoned channels, shallow satellite lakes, and ponds, rather than a uniform open lake. Given that the Shungura Formation is located upstream of the system (on the northern lake coast, near the primary tributary), there may have been an early emergence in this region. In contrast, the basin area likely retained a relatively clear lacustrine environment, as indicated on the map in Figure 2. The G-sup Member records the initial declines in lake level and the subsequent development of a delta, particularly from G-27 (Haesaerts et al., 1983). In Figure 8, lacustrine facies are briefly documented in the Shungura Formation, primarily marking the end of the lake's transgression and the beginning of its regression during the Lorenyang highstand. Specifically, G-24 appears as an exceptional event within the sequence: sedimentological and paleontological data from this level exhibit distinct characteristics, potentially indicating secondary evaporitic facies and mass fish mortality events (Van Damme and Gautier, 1972; Haesaerts et al., 1983). The

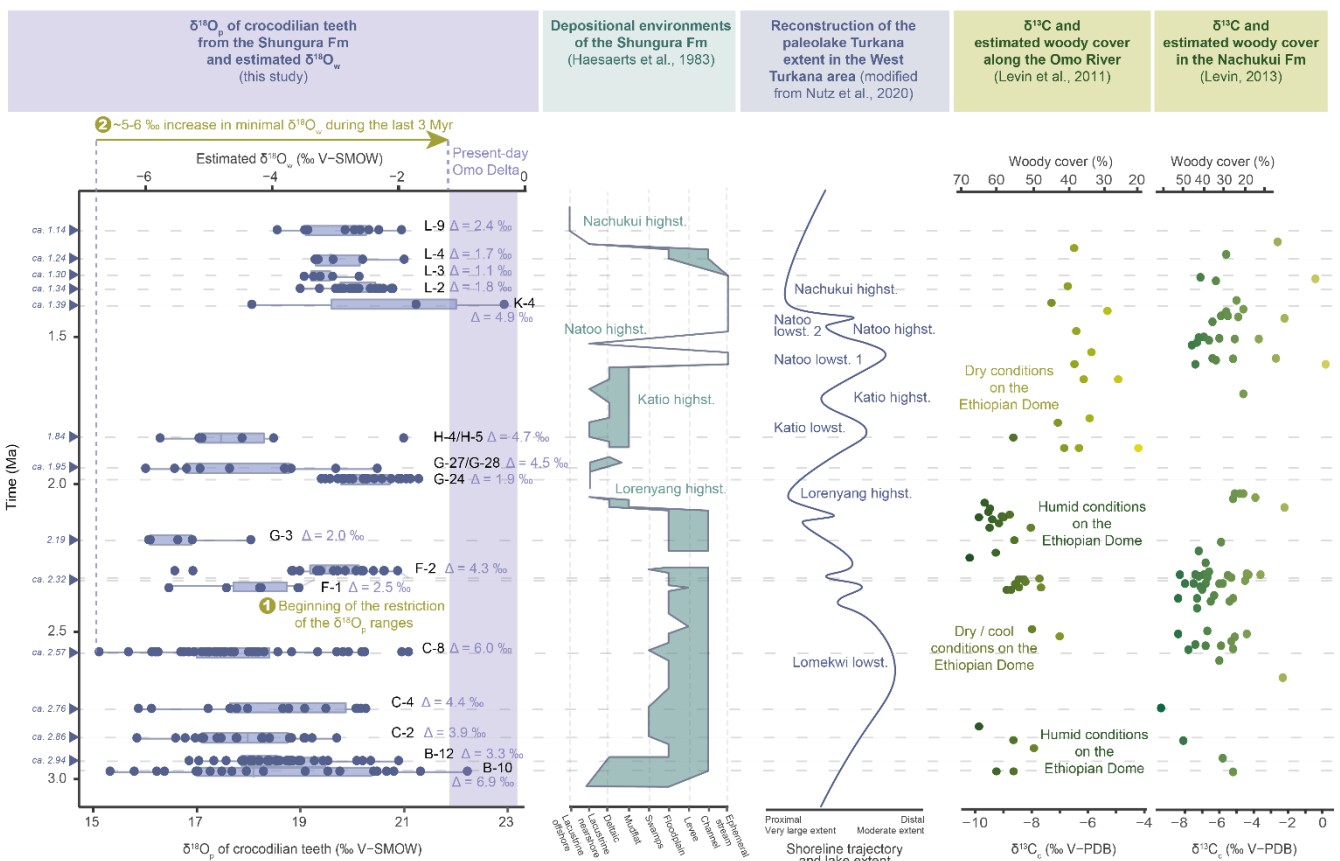

**Figure 8. Comparison of oxygen isotopic composition of crocodilian teeth ($\delta^{18}O_P$) and estimated for surface water ($\delta^{18}O_w$) fluctuations with data on the evolution of sedimentary systems of the Shungura Formation, Lake Turkana level and woody cover in the Shungura Fm.**

inclusion of G-24 within lacustrine layers supports that it may carry a climatic significance, possibly signifying an abrupt arid event. Thus, this could correspond to a particular short event where the water supply is particularly low within a trend of aridification of the Ethiopian Dome. The isotopic data of crocodilians align with this interpretation, as reflected in the strongly positive d $\delta^{18}O_w$, signifying a highly evaporated water body.

The G-27/G-28 and H-4/H-5 units exhibit similar isotopic composition pattern, featuring a wide range of $\delta^{18}O$ values, including both more negative and even some positive values akin to those observed in G-24 (Fig. 8). This suggests that crocodilians had access to at least two distinct types of water bodies during this period: an evaporated lake and its tributary source.

The observed 4 ‰ difference between the isotopic compositions aligns with expectations for a contemporary Ethiopian exoreic lake and its tributary, as documented by Hailemichael et al. (2002, Table 2). This shift in isotopic patterns may illustrate the ongoing regression of the lake during the Katio lowstand, marking the establishment of a deltaic environment. This transition is consistent with previous sedimentological studies (Nutz et al., 2020) (Fig. 8). Notably, the present-day extensive Omo Delta exhibits a diverse range of sub-environments, including active or abandoned distributary channels and interior lakes that are

not directly connected to the open lake. This modern landscape provides valuable insights into the Pleistocene environment (Schuster et al., 2022). With the Shungura Formation situated at the location of the primary tributary to Lake Turkana, specifically the Omo and its delta, the shift from shallow, evaporated lake facies in G-24 towards a deltaic environment was

410 gradual. The crocodilians in the G-27/G-28 and H-4/H-5 units inhabited the interface between the tributary (less evaporated) and the lake (highly evaporated, resembling the conditions in G-24). This regression appears to be linked to climatic factors, potentially induced by a decrease in precipitation over the Ethiopian Dome, as suggested by Nutz et al. (2020).

The three teeth analysed in unit K-4 show divergent values, reflecting the presence of at least two different water bodies accessible to crocodilians, but But this interpretation must be viewed with caution because the number of teeth available for

this unit was low.

Units L-2 to L-4 show particularly high minimal $\delta^{18}O_p$ values, indicating that the crocodilians only had access to one type of water body, evaporated and/or that the $\delta^{18}O$ of precipitation increased. The depositional environments in these units are described as ephemeral streams by Haesaerts et al. (1983), therefore at least seasonally evaporating water bodies could have existed (Fig. 8). The L-9 unit does not appear to be different from the other units analysed in the Member L in terms of the

420 oxygen isotopic composition of crocodilian enamel. Nevertheless, sedimentary studies demonstrate that it was deposited in a lacustrine context (Haesaerts et al., 1983). The distinction in $\delta^{18}O_w$ between ephemeral streams and the lake may not be pronounced when the evaporation conditions of the water bodies, including the ephemeral stream and offshore lake waters, are similar. Alternatively, this similarity might be balanced by a shift in the $\delta^{18}O$ of precipitation, although this is less likely. It is important to note that the geochemical approach used in this study cannot provide a definitive answer to this question.

**4.3.1 Long paleoclimatic trend, the increase in $\delta^{18}O_w$ during the last 3 Myr**

The minimum estimated values of $\delta^{18}O_w$ display an evolution through time, increasing from $-6.5 \pm 2$ ‰ at 2.97 Ma to $-3.9 \pm 2$ ‰ at ca. 1.14 Ma. In general, estimated $\delta^{18}O_w$ values do not, or barely, reach (considering the 2 ‰ error on the estimate) the modern $\delta^{18}O_w$ values of Omo River waters at the delta (between $-1.2$ ‰ and $-0.1$ ‰), and of the Ethiopian lakes (between 1.5 ‰ and 7.2 ‰) (Table 2). Considering minimum $\delta^{18}O_w$ value at 2.97 Ma and comparing it to present-day values, our

reconstructed $\delta^{18}O_w$ data show an increase between $5.3 \pm 2$ ‰ to $6.4 \pm 2$ ‰ (compared to the Omo River) (Table 2). Again, it should be kept in mind that these values represent a maximum estimate of the ancient $\delta^{18}O_w$ values. Therefore, the difference between the ancient and the present values may be greater because the rainwater probably suffered from evaporation before its uptake crocodilian body water. This increase in $\delta^{18}O_w$ is similar to that evidenced by the oxygen isotopic composition of paleosols and fossil molluscs in other Plio-Pleistocene eastern African deposits (Hailemichael et al., 2002; Levin et al., 2004)

and thus requires a regional explanation.

The increase in $\delta^{18}O_w$ cannot be explained by major changes in the topography in the Ethiopian and Kenyan domes and the eastern African Rift (Abbate et al., 2005; Nutz et al., 2017; Ragon et al., 2019), but could be due a rarefaction of rainfalls. The extent of this remains difficult to estimate, as the oxygen isotopic composition of meteoric water ($\delta^{18}O_{mw}$) in eastern Africa is currently not highly correlated with rainfall intensity. Levin et al. (2004) suggest that a 1 ‰ increase in $\delta^{18}O_{mw}$ would reflect

a 32 mm decrease in precipitation, based on GNIP stations in East Central Africa. The data from Table 2 shows a relatively weak correlation between $\delta^{18}O_{mw}$ and the amount of precipitation ($R^2$ = 0.21), suggesting that a 1 ‰ increase in $\delta^{18}O_{mw}$ could be associated with a 160 mm decrease in precipitation. In the case of the Ethiopian Dome, with $\delta^{18}O_{mw}$ values at least 5.3 ‰ lower than today's, Addis Ababa would then have received 870 mm more precipitation in the late Pliocene, thus reaching about 2050 mm/year of precipitation. This amount is close to that recorded now in the equatorial forest of Congo (ex. Goma GNIP station, IAEA, 2023, Table 2). If the Pliocene climatic regime were similar to the present day's one, then higher sea surface temperatures in the Indian Ocean could have led to increased precipitation in eastern Africa at that time. This hypothesis is consistent with the progressive global cooling observed in the last million years (Zachos et al., 2001). Although there is no direct evidence for the existence of a rainforest on the Ethiopian Dome during the Pliocene, paleobotanical studies reveal a period of drastic rainforest retreat between 3.5 and 2.5 Ma, and probably before the first hominins in the region (Bonnefille, 2010), and the installation of an assemblage of mountain forest plants on the Ethiopian Dome, during global cooling (Bonnefille, 1983).

On the other hand, the sources and trajectories of precipitation play a major role in the variation of $\delta^{18}O_{mw}$ in eastern Africa. Changes in the location of the Intertropical Convergence Zone (ITCZ) or Congo Air Boundary (CAB) are more likely to strongly impact the variation of the isotopic composition of the eastern African meteoric water, potentially exceeding 3 ‰ (Kebede and Travi, 2012). With a warmer Indian Ocean, the ITCZ would have been expected to move northward, and the Ethiopian Dome would have received a larger proportion of Indian Summer Monsoon rains, which have lower $\delta^{18}O_{mw}$ values than the West African Monsoon. Alternatively, a more westerly location of the Congo Air Boundary in Ethiopia during the Plio-Pleistocene could have lowered the $\delta^{18}O_{mw}$ values up to the extent recorded 2.97 Ma in the Lower Omo Valley (Kebede and Travi, 2012; Levin et al., 2009).

**Conclusions**

Given their low metabolism and aquatic ecology, crocodilians stand out as excellent recorders of environmental conditions, with their $\delta^{18}O_p$ values closely linked to that of the water they live in during teeth formation. Combining knowledge of the physiology, ecology and development of crocodilians is of crucial importance to further interpret their oxygen isotopic signal: it allows indirectly to gain insights into the local diversity of aquatic environments, local evaporation and $\delta^{18}O$ values of meteoric waters. This approach opens a new understanding of past freshwater environments and is complementary to sedimentological studies and terrestrial proxies. Given the uncertainty surrounding the season and duration of tooth mineralization and their potential impact on isotopic interpretations, further investigation on this issue could lead to a refined this model. It could elucidate whether the observed diversity of freshwater environments within a stratigraphic unit represents year-round availability, season-specific occurrences, or variations across different seasons. The study of paleohydrological variability should be conducted at the stratigraphic unit scale to fully capture its potential, and potentially within units based

on lateral facies variations. A broader-scale study, such as that of the Shungura Formation members, would not provide such detailed insights.

The new data set suggests that between 2.97 Ma and ca. 2.57 Ma, the Lower Omo Valley hosted a great diversity of water body types, with different water sources or evaporative states. From ca. 2.32 Ma, the diversity of freshwater environments decreased drastically to only one or two types of water bodies at each stratigraphic unit, probably due to aridification that caused some water bodies to either disappear or to become further separated and isolated, making them inaccessible to local crocodilians. This freshwater environment restriction prevailed in the rest of the Shungura Fm, except at ca. 1.95 Ma and 1.84 Ma: the lake level decreased giving way to a deltaic environment in the Shungura Fm with a high diversity of water bodies.

The $\delta^{18}O_p$ of crocodilian teeth also carries valuable information about regional paleoenvironments but cannot help identifying the exact cause between different potential factors (origin of water, rainfall amount, topography, temperature, etc.). This new dataset highlights a large increase in $\delta^{18}O_w$ values, between 5.3 ‰ and 6.4 ‰, from the Pliocene to the present, consistent with other studies in eastern Africa (Hailemichael et al., 2002; Levin et al., 2004). This increase is likely to have resulted from the migration of convergence zones of low air streams affecting the distribution of the monsoons affecting the $\delta^{18}O_w$ in addition to the resulting scarcity of rainfalls over the Ethiopian Dome. The data point to a main shift in $\delta^{18}O_w$ between 1.84 Ma and ca. 1.34 Ma, slightly later than suggested by studies on paleosols (Levin et al., 2011). While several terrestrial and aquatic proxies indicate aridification in the Turkana Depression between 2.8 Ma and 2.5 Ma, the $\delta^{18}O_p$ of crocodilian teeth points to the stability of the local diversity of aquatic environments during this period. Crocodilian teeth are also abundant in continental series and widely distributed across the globe, notably in the Neogene strata of Africa. Although adapted in the context of the Shungura Fm, the proposed interpretation model can be generalized to comparable paleontological and archaeological sites, especially in other sites in the Turkana Depression but will require sampling as detailed as that done by OGRE in the Shungura Fm. For optimal use of this approach, particularly along another stratigraphic sequence in the intertropical zone, it is necessary to ensure that temperature changes are limited and do not significantly affect the body temperature of crocodilians. For the same purpose, it is necessary to compare the same taxa throughout the sequence to avoid interspecific variation. Better control of the potential phylogeny-dependent variation of $\delta^{18}O_p$ in the present dataset would benefit from a study on the systematics and temporal distribution of the different crocodilian taxa in the Shungura Fm. Once the preservation of the original isotopic signal of the crocodilian fossil tooth enamel is ensured and the physiological, ecological, and environmental parts of the signal identified, then the $\delta^{18}O_p$ could be used to characterize the aquatic environments and their diversity over time in many other continental intertropical contexts.

**Supplement**

The supplement related to this article is available online at https://zenodo.org/records/10084380.

## Author contributions

AG, EP, GG, JRB, and OO designed the study. EP, GG, and OO provided supervision. JRB supervised fieldwork and specimen sampling, involving AN, MS, and OO. AG and EP performed enamel sampling and preparation in the lab. EP and MMJ performed the stable isotope analysis and data treatment. AE, AG, and JRB built the age model for the Shungura Fm. AG was responsible for statistical analysis, data curation, and visualisation, and wrote the paper with contributions from all the co-authors.

## Competing interests

The contact author has declared that none of the authors has any competing interests.

## Acknowledgements

The authors are indebted to all the people who contributed to the fieldwork in the Shungura Formation as part of the Omo Group Research Expedition (OGRE, PI: Jean-Renaud Boisserie). OGRE corresponding fieldwork seasons were funded notably by the UMR 7262 PALEVOPRIM (CNRS and University of Poitiers), the French Ministry of Europe and Foreign Affairs, the CNRS (SEEG Shungura), the French National Agency for Research, the Fyssen Foundation, and were logistically supported by the UAR 3137 CFEE (CNRS and Ministry of Europe and Foreign Affairs). Fieldwork, sampling material exportation, and destructive analyses of fossil remains were authorized by the Ethiopian Heritage Authority (previously ARCCH) within the framework of the OGRE research permit, and greatly facilitated by its staff members. The authors also thank Sophie Hemry and Guillaume Di Bez who prepared 80% of the samples for phosphates during their master thesis. The authors are grateful to Ivan Jovovic (UMR 5561 Biogéosciences, CNRS and University of Burgundy) who analysed part of the enamel carbonates sample. This work is part of the PhD thesis of AG, who received financial assistance from the UMR 7262 (PALEVOPRIM, CNRS and University of Poitiers), the ASAP project (project n°210724, Nouvelle-Aquitaine region, France, PI: Olga Otero) and the French Ministry of Higher Education, Research and Innovation.

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
