# Peer review of "Stable oxygen isotopes of crocodilian tooth enamel allow tracking Plio-Pleistocene evolution of freshwater environments and climate in the Shungura Formation (Turkana Depression, Ethiopia)."

_Biogeosciences, 2023_

## Author Response (AR1)

Laboratoire Paléontologie Evolution Paléoécosystèmes Paléoprimatologie
UMR 7262 CNRS – Université de Poitiers, France

**Axelle GARDIN**
Postdoctoral Researcher

Poitiers, November 08, 2023

Dear Co-editors-in-chief,

Enclosed please find our revised manuscript bg-2023-125, entitled 'Stable oxygen isotopes of crocodilian tooth enamel allow tracking Plio-Pleistocene evolution of freshwater environments and climate in the Shungura Formation (Turkana Depression, Ethiopia)'.

We appreciate the time and effort that the two referees dedicated to providing feedback on our manuscript and we believe this certainly will help clarify and improve some points of this paper. We have incorporated most of the comments and suggestions proposed by the referees, and you will be able to follow the modifications and revised sections in the manuscript. We have posted the additional material on Zenodo: https://zenodo.org/records/10084380.

In the paragraphs below, we explain the main changes we made on the manuscript following the comments of the referees:

**\*\*\*\*\***

**Anonymous Referee #1, 26 Sep 2023**

Dear editor,

The manuscript by Gardin et al. presents an interesting study on the use of the stable oxygen isotopic composition of crocodilian tooth enamel to assess changes in hydrological conditions. The general idea is that if crocodiles have access to a larger array of aquatic environments (lakes, rivers, swamps), their teeth show a larger spread in isotopic data, whereas if they have limited access, for example only to one river, the isotopic composition of their teeth will show a narrower distribution. Therefore, the isotopic composition of crocodilian tooth enamel, and spread thereof, provides some (indirect) insights in the hydrological conditions of a region, with a larger array of accessible aquatic environments during wetter periods, and less accessible aquatic environments during dryer periods.

This application of the stable oxygen isotopic composition of crocodilian tooth is relatively new, and the authors use the well-dated deposits of the Shungura Formation as a testcase. Their results indeed provide some first insights in the changes in aquatic environment accessibility through the succession, with some of the highstand-periods marked by a larger spread of data, and some of the lowstand-periods marked by narrower ranges. In other intervals, the data is less straightforward to interpret, as can be expected for these kind of relatively complex proxies.

Dear referee,

We are grateful for the time and great effort you dedicated to carefully reviewing our manuscript and for your interesting and detailed comments. We understand your concerns and have improved the manuscript to take them into account and gain clarity. We have corrected all the minor points you raised in the revised version, and we answered your major concerns (it italics) that needed explanation in this reply.

Université de Poitiers - UFR SFA
Bât. B35 - TSA 51106 - 6, rue Michel Brunet - 86073 POITIERS Cedex 9 - France
http://palevoprim.labo.univ-poitiers.fr/ - axelle.gardin@univ-poitiers.fr

[Figure]

[Figure]

Laboratoire Paléontologie Evolution Paléoécosystèmes Paléoprimatologie
UMR 7262 CNRS – Université de Poitiers, France

**Axelle GARDIN**
Postdoctoral Researcher

In parallel, we received information regarding the crocodilian taxa identified at Shungura from a peer who read the preprint. The genus *Rimasuchus* is not present in the Turkana Depression, and the largest crocodilians may belong to the species *Crocodylus thorbjarnarsoni*. We make the correction in the manuscript.

In general, the manuscript is well written and nicely illustrated. It presents an interesting contribution, about a novel approach that might have future applicatons in other successions. However, more detailed information about the biology behind crocodilian teeth growth would have been valuable, to better understand the proxy. For example, the authors discuss that tooth formation in crocodiles occurs over several months. Is there any information about over which months this growth occurs? Is this random throughout the year, or can it be that tooth growth was for example stronger in the wet season, as opposed to the dry season? Given the strong seasonal rainfall, this might have significant consequences for the isotopic composition of the teeth. Continuing on this, if tooth formation of crocs occurs over several months, the entire tooth will likely reflect only one season. Hence, the strongest seasonal variation would be expected between teeth from the same locality, rather than within one tooth, right? Is the number of teeth analyses per level sufficient to representative of the entire year? Or could it be that some intervals are characterized by a bias towards one season?

We understand your concerns about the number of samples per level. We checked in advance the minimum number of samples necessary to have stability of the isotopic signal, but we acknowledge that these details are missing in the manuscript. We, therefore, specify in the revised Material and Method: "To determine the minimum number of teeth required for a reliable representation of isotopic amplitude, an approach based on data from level C-8 was employed due to the number of teeth analysed and the large range of the values. Applying the slope break criterion established a threshold of 6 teeth as optimal to stabilize the amplitude measurement. However, in 5 levels, fewer teeth were available, and it is acknowledged that this requires increased caution in interpreting these results.". In addition to this paragraph, we specifically add in the revised Discussion for the levels affected by under-sampling that the interpretation should be more cautious.

Here and there, some additional information would be usefull as well. For example: what is the average size of the teeth studied? The average height of the completely preserved teeth is 18.2 mm for the rounded teeth and 30.0 mm for the pointed ones. We add this clarification in the revised manuscript. Finally, the key figure of this manuscript (Fig 8) could be improved a bit to better highlight the results of this study. We improve Figure 8 in this sense. See below for a complete list with comments and suggestions. Taken together, I would recommend this manuscript to be accepted following some revisions and clarifications.

Comments and suggestions

Line 69: here the authors discuss that tooth formation in crocodiles occurs over several months. Is there any information about over which months this growth occurs? Is this random throughout the year, or can it be that tooth growth was for example stronger in the wet season, as opposed to the dry season? Given the strong seasonal rainfall, this might have significant consequences for the isotopic composition of the teeth, right? Line 151: to continue on this matter: if tooth formation of crocs occurs over several months (line 69), the entire tooth will

Université de Poitiers - UFR SFA
Bât. B35 - TSA 51106 - 6, rue Michel Brunet - 86073 POITIERS Cedex 9 - France
http://palevoprim.labo.univ-poitiers.fr/ - axelle.gardin@univ-poitiers.fr

[Figure]

[Figure]

Laboratoire Paléontologie Evolution Paléoécosystèmes Paléoprimatologie
UMR 7262 CNRS – Université de Poitiers, France

**Axelle GARDIN**
Postdoctoral Researcher

likely reflect only one season. Hence, the strongest seasonal variation would be expected between teeth from the same locality, rather than within one tooth, right?

We acknowledge the importance of the question regarding the dental development of crocodilians and its potential impact on the interpretation of isotopic signals. The seasonality of crocodilian tooth mineralisation is an aspect to consider when analysing our results.

Duration of Tooth Formation:

Existing literature is unfortunately not extensive on this matter. According to previous studies, tooth replacement in crocodilians, especially in alligators, occurs approximately once a year, with a range of eight to sixteen months, depending on the tooth's position in the dental row (Edmund, 1962).

Other studies, such as those by Erickson (1992, 1996b), based on dentin increment counting, estimate tooth formation times in both modern and fossil crocodilians to be between 83 and 285 days and seem to increase with age and body size.

We are currently not able to definitively state whether tooth formation occurs during a particular season, and it is important to note that crocodilians may estivate or hibernate, which can affect somatic and maybe dental growth (Chabreck and Joanen, 1979; Huton, 1987; Taplin, 1988).

Impact on Isotopic Signal:

In the case of dental growth spread over several months during different seasons, seasonal variations in the environment can be attenuated in dental enamel compared to the actual amplitude of drinking water, as observed in mammals.

We consider the interannual variability of the isotopic signal in the Turkana Depression, which can be influenced by different types of aquatic environments and water mixing, as evidenced by mollusc shells in the Omo delta (range of 6 ‰) Omo River (range of 2.5 ‰); Turkana Lake (negligible fluctuations) (Vonhof et al., 2013).

While we cannot currently determine the season and duration of tooth mineralization definitively, we compare the same isotopic signal throughout the stratigraphic sequence by ensuring that the minimum number of teeth is reached for a reliable representation of its amplitude. We mention this in the revised manuscript and review the interpretations of the levels with the fewest teeth more cautiously. We observe a significant change in aquatic landscapes over time, but it is difficult to say whether this is on an annual or seasonal scale within each level.

We will incorporate this information into the discussion and limits of our results to better contextualize our conclusions.

Line 196: "slightly different sedimentary facies" is a bit vague. Can the authors explain in what way the sedimentary facies differed?

This is lateral variation, between relatively more silty, more stratified levels in spot a, and something a bit coarser, with less structure in spot b. We include this information in the revised manuscript.

[Figure]

Laboratoire Paléontologie Evolution Paléoécosystèmes Paléoprimatologie
UMR 7262 CNRS – Université de Poitiers, France

**Axelle GARDIN**
Postdoctoral Researcher

Lines 225-226 "….and the good match of values to other geochemical studies dealing with modern and ancient crocodilians.": Please provide the corresponding references here.

We added references in the revised manuscript.

Line 250 "especially for large individuals":  what is considered "large" here? Line 252: again, what is large here? Can the authors give an indication of the average size of the teeth used in this study? So far only the minimum size was discusses (1 cm). A tooth of 1 cm would not come from a 'large' croc..

We specify in the revision that large individuals are those with a weight of 100 kg or more, according to the article by Smith (1979). The average height of the completely preserved teeth is 18.2 mm for the stout (rounded) teeth and 30.0 mm for the pointed ones. We add this clarification in the revised manuscript.

We estimate the length of a 100 kg crocodile at ~2.8 m. With a rough calculation from photos of living specimens or skulls, we can deduce that for a total length of 2.8 meters, the crown of the teeth of C. niloticus could measure between 7 mm (molariform) and 38 mm (caniniform), and between 7 mm and 19 mm for Mecistops.

Except for a 6 mm molariform tooth, all other molariform teeth have a crown height greater than 1 cm. For other stout teeth, they are all at least 1 cm long, and 3 pointed teeth have a height of less than 18 mm. We add a column in the supplementary data and conclude that most teeth analysed belong to large specimens, and we can roughly estimate the minimum size at 2 meters but most of the individuals may have reached more than 2.8 m.

Lines 253-255 "Given their capacity for thermoregulation, crocodilians occupying the same environment probably do not always have the same body temperature.":  I don't exactly follow here. Wouldn't all those individuals of one species generally show more or less the same behavior? After all, all individuals will have the same tendence for heat-seeking or -avoidance, by land-water movements. Or are do the authors suggest/mean to say that indivudals do not always have the same body temperature through their life? Perhaps this sentence can be clarified a bit further, also more clearly explaining the reasoning behind this statement.

We clarify this sentence as follows: "Given the thermoregulation abilities of crocodilians, depending on the individuals and their behaviour and environment, the body temperature can be slightly different between individuals occupying the same water body, and possibly greater between two populations occupying different aquatic environments (Amiot and al., 2007). The body temperature of crocodilians varies mainly in a range of 10°C, which approximately corresponds to a difference of 2.3 ‰ in $\delta18Op$ (Longinelli and Nuti, 1973; Kolodny et al., 1983; Pucéat et al., 2010) (Fig. 6).". By taking this margin of 2.3% we want to take precautions for possible temperature differences between individuals (indeed because of behaviour, age and size) and populations.

Lines 304-305 "Calculation of $\delta18Ow$ values from $\delta18Op$ of Shungura fossil crocodilian enamel is based on the fractionation equation of Amiot et al. (2007).": I think it would improve the readibility of this manuscript, if this information was provided slightly earlier in the text.

Université de Poitiers - UFR SFA
Bât. B35 - TSA 51106  -  6, rue Michel Brunet  -  86073 POITIERS Cedex 9 - France
http://palevoprim.labo.univ-poitiers.fr/   -   axelle.gardin@univ-poitiers.fr

[Figure]

[Figure]

Laboratoire Paléontologie Evolution Paléoécosystèmes Paléoprimatologie
UMR 7262 CNRS – Université de Poitiers, France

**Axelle GARDIN**
Postdoctoral Researcher

We add this information also in Material and Method, section 2.4.

Lines 340-341 "This could be explained by less evaporation or less precipitation in the lower Omo Valley during the 2.8-2.5 Ma period but there was less water coming from the Ethiopian Dome.": this sentence reads a bit odd. Perhaps reformulate?

We reformulate: "These $\delta^{18}O_p$ values do not indicate aridification affecting freshwater environments between 2.8 Ma and 2.5 Ma (Members B and C). This contrasts with changes observed in the aquatic and terrestrial ecosystems of the eastern African Rift: such as lake level drop, landscape opening, and aridification (e.g., Maslin and Trauth, 2009; Trauth et al., 2009, 2021; Levin et al., 2011; Negash et al., 2015, 2020; Blondel et al., 2018, 2022; Nutz et al., 2020) (Fig. 8). Interestingly, the $\delta^{13}C$ of soil carbonates increases, thus indicating aridification in the Shungura Fm but not in the Nachukui Fm in the West Turkana area (Levin et al., 2011; Levin, 2013; Nutz et al., 2020) (Fig. 8). Nutz et al. explain (2020) that the $\delta^{13}C$ record in the Shungura Fm reflects climatic conditions upstream on the Ethiopian Dome, and that of Nachukui Fm reflects more local and regional climatic conditions. The $\delta^{13}C$ of the paleosols would therefore indicate a less significant supply of water from the Ethiopian Dome towards the Lower Omo Valley. On the other hand, the hydrological conditions remain stable in the western part of the Turkana Depression between 2.8 Ma and 2.5 Ma, thus possibly preserving a certain diversity of aquatic environments in the surroundings."

Line 357: Why does the beginning of a regression immediately lead to evaporative facies? Perhaps the authors could explain a little bit more about what is already known/discussed about this event? To the less-informed reader, it strikes as odd that a level very close to a highstand is characterized by evaporative conditions, based on the isotopic data from the crocs perhaps even among the most extreme of the succession? Especially the level above this, which should represent the continuation of this regression towards the Katio Lowstand, is characterized by a larger spread in data and lighter values. This seems counterintiuive...

The lake reached its maximum extension between approximately 2.10 Ma and 1.90 Ma, which brings it up to the Shungura Fm which shows lacustrine facies. According to Nutz et al. (2020) the regression begins around 2 Ma (G-24 is dated approximately at 1.98 Ma), when we see a migration of the shoreline, towards the basin and delta facies reappearing progressively in G-sup Member. So, we can imagine more segmented aquatic environments (distributary channels, abandoned channels, shallow satellite lakes, ponds, ...) than the simple open lake.

Since the Shungura Fm is located upstream of the system (northern coast of the lake, near the main tributary), there may be an early emersion in this sector while further towards the basin there is still clear lake (see map on Fig. 2). It records the first drops in lake level in G-sup Member and later the installation of a delta later from G-27 (Haesaerts et al., 1983). In Figure 8, it can be seen that lacustrine facies are recorded briefly at Shungura Fm and only at the end of the transgression and the beginning of the regression of the la during the Lorenyang highstand. Our results only support previous studies: in fact, level G-24 seems to stand out as a particular event within the sequence. The sedimentological and paleontological data from level G-24 are very particular and indicate (secondary?) evaporitic facies and mass fish death events (Heinzelin 1983). The isotopic data of crocodilians is congruent with this because the d18O of the water is very positive reflecting a strongly evaporated waterbody. G14 could therefore have

Université de Poitiers - UFR SFA
Bât. B35 - TSA 51106 - 6, rue Michel Brunet - 86073 POITIERS Cedex 9 - France
http://palevoprim.labo.univ-poitiers.fr/ - axelle.gardin@univ-poitiers.fr

a climatic meaning (i.e., abrupt arid event) or simply a meaning of the internal functioning of the system (i.e., abandoned basins where salt precipitates). But as G14 is intercalated in lacustrine layers, we are betting instead on option 1. In the case of option 2, G14 would be intercalated in fluvial/deltaic. Thus, this could correspond to a particular short event where the water supply is particularly low within a trend of aridification of the Ethiopian Dome. This part is reformulated in the revised manuscript to avoid these misunderstandings.

Lines 362-264 "This change in Isotopic pattern illustrates the progressing regression of the lake (Katio lowstand) and the establishment of a deltaic environment, consistent with previous sedimentological studies": I do not follow here. So the change from a highly evaporative water body, to an environment with at least two water bodies (evaporative water body and tribuatries) indicates a progressing regression? There are actually more datapoints with isotopically lighter values in G27/G28 than in G24. Wouldn't that be indicative of more freshwater environments? This would be inconsistent with a progressing regression from G24 to G27/G28? Or is it a transition from lake to deltaic, because the lake level drops beyond the deposition site? Please clarify the line of reasoning here.

As the Shungura Fm is located at the location of the main tributary of Lake Turkana, the Omo and its delta, during regression the shallow lake facies in G-24 gradually give way to a deltaic environment: the crocodilians of the levels G-27/G-28 and H-4/H-5 are found at the interface between the tributary (little evaporated) and the lake (highly evaporated, with values similar to G-24). The regression would have a climatic control induced by a decrease in precipitation over the Ethiopian Dome (Nutz et al., 2020). It can be assumed in G-24 that the crocodiles remained in this evaporated and relatively isolated lake during an abrupt arid event. This part is reformulated in the revised manuscript to avoid these misunderstandings.

Lines 374-374: It might not be the most likely scenario, that the difference in d18O in lake and streams between L2-L4 and L9 would be exactly compensated by a change in the d18O of precipitation, right? And if this is a likely scenario, it remains an issue for this novel proxy…

Indeed, with our isotopic data, it is not possible to resolve the question of controls on the origin of water or evaporation. Sedimentary data indicate different depositional environments but are both subject to high evaporation. We add a sentence to draw attention to this issue. This is a point we developed in the conclusion section.

Lines 397-398: is there any other evidence of the existence of tropical rain forests on the Ethiopian Dome during the Pliocene?

There is no direct evidence for the Ethiopian Dome. However, paleobotanical studies reveal a period of drastic rainforest retreat between 3.5 and 2.5 Ma, and probably before the first hominins in the region (Bonnefille, 2010), and the installation of an assemblage of mountain forest plants on the Ethiopian Dome, during global cooling (Bonnefille, 1983).

Lines 399-402: To me, this far out seems the most likely explanation.

Indeed, but unfortunately our data does not allow us to decide directly.

[Figure]

Laboratoire Paléontologie Evolution Paléoécosystèmes Paléoprimatologie
UMR 7262 CNRS – Université de Poitiers, France

**Axelle GARDIN**
Postdoctoral Researcher

Line 411: I would add the word "indirectly" to this sentence: "….of aquatic environments, local evaporation and indirectly into δ18O values of meteoric waters"

We corrected "it allows to indirectly gain insights into the local diversity of aquatic environments, local evaporation and δ18O values of meteoric waters".

Lines 413-415 "It is noteworthy…..Shungura Formation.": This sentence feels more as if it belongs in the Discussion section, rather than in the conclusions.

We agree with your comment, we can put this sentence in the discussion, but we also find it important to remember it in the conclusions, because it is an important point to take into account for applications to other contexts.

Lines 434-436 "For optimal use of this approach, particularly along a stratigraphic sequence, it is necessary to ensure that temperature changes are limited and do not significantly affect the body temperature of crocodilians.": I guess this means that this proxy would be mostly applicable in tropical regions? Note that a there are/were consiserable populations of crocodilians in (subtropical) regions that do/did show a considerable temperature seasonality.

Yes, you are absolutely right, we forgot to specify it in this sentence.

Figure 8: please quantify the range in d18Op values per level. After all, this is considered one of the main informative signals? Also: the Shungura Fm. is characterized by a rather precise age control. This is not really depicted in the vertical (time) axis, which has tick marks every half a million years! For readability, please increase the number of tick marks on the vertical axis, so one can better assess the ages of the various studied levels.

We modified the figure accordingly.
* * *
**Anonymous Referee #2, 27 Sep 2023**

This manuscript by Gardin et al unlocks a new stable isotope proxy record based on fossil crocodile teeth for the Plio-Pleistocene Shungura record in the East African Rift. There is potential in this record, as the d18O values of tooth enamel phosphate are mostly controlled by the d18O values of the water the crocodiles lived in. Furthermore, tooth enamel is one of the most diagenesis-resistant materials in the fossil record, increasing the likelihood that the croc teeth have contained the original isotope signal of the water they lived in.

For the interpretation of the record the authors use the range of isotope data of teeth collected from discrete stratigraphic intervals and interpret that in terms of the range of different aquatic environments available to the crocs during that time interval. Likely the most important parameter in determining that range is the level of evaporation in these aquatic environments, a parameter that will be strongly influenced by the depositional setting, which was quite dynamic in the Shungura fm, switching between largely riverine and lacustrine, dependent upon the lake level changes of the nearby paleo-lake Turkana

Université de Poitiers - UFR SFA
Bât. B35 - TSA 51106  -  6, rue Michel Brunet  -  86073 POITIERS Cedex 9 - France
http://palevoprim.labo.univ-poitiers.fr/   -   axelle.gardin@univ-poitiers.fr

[Figure]

[Figure]

Laboratoire Paléontologie Evolution Paléoécosystèmes Paléoprimatologie
UMR 7262 CNRS – Université de Poitiers, France

**Axelle GARDIN**
Postdoctoral Researcher

In the ms the interpretation of crocodile tooth isotope ratios is outlined in relation to these changes, and some scenarios are presented for the longer-term changes underlying the croc tooth isotope pattern through time.

I find this an original and interesting paper, but have some discussion points and questions concerning:

- Your approach approach towards proving the good preservation of enamel
- The isotope ranges of the collected teeth in relation to the depositional setting
- The long paleoclimatological trend and the rather low d18Ow values calculated for the older part of the sequence

We would like to express our gratitude for the time dedicated to reviewing our manuscript and for bringing up these inquiries, which contributed to the improvement of our research. The detailed questions and comments that were raised will be thoroughly addressed in the subsequent sections of our response.

In parallel, we received information regarding the crocodilian taxa identified at Shungura from a peer who read the preprint. The genus *Rimasuchus* is not present in the Turkana Depression, and the largest crocodilians may belong to the species *Crocodylus thorbjarnarsoni*. We make the correction in the manuscript.

1) Preservation state of the material:

I understand you have not pre-treated samples (by leaching or otherwise) before doing the analyses. I would generally not object to that as pretreatment often enough does not really improve the data for samples that may have some diagenetic overprint. What you then do to investigate the preservation state is cross-plot the d18Op with d18Oc (phosphate vs structural carbonate of the tooth). This has been done before on modern tooth material, which generally leads to a relatively clear correlation with a constant d18O offset. In the plot that you produce in this context (Fig 5), you not only show modern material, but also (Cretaceous) fossil material that clearly show a different pattern (and slope), and then -if I got this right- you reason that because your own data fall in between the two other datasets, the preservation of your material should be OK. Frankly, I don't follow why that pattern indicates your data is OK? I would personally be worried when my fossil data show a different slope than the modern dataset, and comparing to another fossils dataset that shows yet another pattern does not directly make sense to me. Still, I am rather willing to believe that your d18Op data are OK, simply because enamel phosphate is so difficult to alter isotopically. It is much more likely that, if there is a diagenetic overprint that contributes to the patterns in Fig 5, you should expect it to be in the d18Oc (structural carbonate), that more commonly incorporates a diagenetic component, particularly if you have not pre-treated the material. I'd be interested to hear your opinion on this.

First, we want to express our agreement with you regarding the significance of basing our interpretations on enamel phosphates, which are known to be more resistant to diagenesis. However, we also wanted to include carbonate data for comparison, as it allows for cross-referencing potential preservation of the signal.

Université de Poitiers - UFR SFA
Bât. B35 - TSA 51106 - 6, rue Michel Brunet - 86073 POITIERS Cedex 9 - France
http://palevoprim.labo.univ-poitiers.fr/ - axelle.gardin@univ-poitiers.fr

[Figure]

[Figure]

Laboratoire Paléontologie Evolution Paléoécosystèmes Paléoprimatologie
UMR 7262 CNRS – Université de Poitiers, France

**Axelle GARDIN**
Postdoctoral Researcher

To our knowledge, there are no dual δ18O data on both carbonates and phosphates in the apatite of modern crocodilians (or other sauropsids), which could potentially differ from those of mammals. Therefore, we chose linear relationships established for modern mammals and fossil vertebrates at least partially preserved.

Amiot et al. (2010) described that fossil data "roughly parallels the oxygen isotope fractionation line established for extant mammals", without values homogenisation, and these values exhibit a linear relationship. These are the arguments they used to consider at least partial preservation of the original signal in their fossils. Following the same approach, we demonstrate at least partial preservation of the original signal by showing the linear relationship in our own data, which is reassuring. By comparing our data to the other two datasets, our intention was to see how our data aligns with biological material. While an exact correspondence can be challenging due to various factors, including the evolutionary history of the different clades, our data fall within the expected range for values considered "at least partially preserved" and therefore reasonably interpretable. This was done to demonstrate that our data do not exhibit extreme deviations from expectations, which could suggest a severe diagenetic overprint.

We will clarify our approach in the manuscript to provide a more comprehensive explanation of our reasoning.

2) isotope ranges in relation to sedimentological context:

You essentially use the d18O data to reconstruct the different water bodies (depositional facies), whereby the range of isotope ratios from all teeth in a specific interval indicate the range of environments accessible to the crocs. Since the sedimentological facies can change rapidly, both spatially, and temporally (seasonally!), a croc living in a river can rather easily move between the highly evaporated water of, for example, a floodplain lake or an oxbow lake, and the low-evaporation water of the main river. So, even if you "only" have the river depositional environment available, the crocs likely still had easy access to sources of evaporated water.

Indeed, our theoretical interpretation model is based on this hypothesis. The teeth found in the Shungura Fm. could have belonged to individuals who formed their teeth in the local waterbody which gives its depositional environment, or to individuals who come from another nearby waterbody in which it formed their teeth and will come to Shungura where it shed it teeth (so we don't have access to the depositional environment of this nearby waterbody in which it formed its teeth).

As long as you're not in a large terminal lake, I would personally expect the teeth to potentially show a wide range of oxygen isotope values, from fully riverine to rather evaporated settings, provided you collected enough teeth to be sure that you cover the full range of a specific area and time interval. I find it difficult to conceptualise a situation in which a croc has only access to waters with a narrow isotope range. In some of the sections where you have a rather narrow range in isotope values (for example G-3 in figure 8), I get the impression that you may not have enough samples to be sure that that you cover the entire range? In other words: Are you statistically sure that the narrower isotope ranges that you show in figure 8 are not at least in part determined by undersampling?

Université de Poitiers - UFR SFA
Bât. B35 - TSA 51106 - 6, rue Michel Brunet - 86073 POITIERS Cedex 9 - France
http://palevoprim.labo.univ-poitiers.fr/ - axelle.gardin@univ-poitiers.fr

[Figure]

[Figure]

Laboratoire Paléontologie Evolution Paléoécosystèmes Paléoprimatologie
UMR 7262 CNRS – Université de Poitiers, France

**Axelle GARDIN**
Postdoctoral Researcher

We understand your concerns about the number of samples per level. We checked in advance the minimum number of samples necessary to have stability of the isotopic signal, but we acknowledge that these details are missing in the manuscript. We, therefore, specify in the revised Material and Method: "To determine the minimum number of teeth required for a reliable representation of isotopic amplitude, an approach based on data from level C-8 was employed due to the number of teeth analysed and the large range of the values. Applying the slope break criterion established a threshold of 6 teeth as optimal to stabilize the amplitude measurement. However, in 5 levels, fewer teeth were available, and it is acknowledged that this requires increased caution in interpreting these results.". In addition to this paragraph, we specifically add in the revised Discussion for the levels affected by under sampling that the interpretation should be more cautious.

3) long paleoclimatological trend

While you have this "noise" of the different water types (and temperature uncertainty) determining relatively wide isotope ranges per unit, I agree that when you focus on the lowest isotope values of each unit you seem to have a long-term signal that roughly goes along with the general long aridification trend known to occur in this part of Africa (as also visible in the Levin et al tree-cover data for the Shungura fm in your fig 8).

The low d18Ow values of your older units, to me are absolutely the most interesting outcome of your study, because at the lowest reconstructed d18Ow values of approximately -6 permille VSMOW you are far away from any source water which flags considerable changes through time in the hydrological system of the Rift Valley. Considering the option of increased (monsoonal) influence from the Indian Ocean for the older samples (as you do) to me indeed seems the most logical solution here. To be frank, this concept of the long-term climate trend in your dataset holds more allure for me than the reconstructions of freshwater environments using the differences in the d18O ranges between stratigraphical units, but that may be a matter of personal taste.

We understand your opinion on this point. For us both results are important, but we agree that the long paleoclimatic trend is masked by the interpretation of the diversity of aquatic environments in the current state of the manuscript. We propose to make the two results appear more distinctly and equally in the discussion and the figures.

I further have some suggestions for smaller changes and clarifications:

Line 27-29: This sentence is unclear to me. Can you please rephrase? "Contrary to some conclusions based on terrestrial proxies, the δ18Op of crocodilian teeth does highlight any major change affecting aquatic environments, rather pointing to stability of these environments between 2.97 Ma and ca. 2.57 Ma."

We rephrase: "While several terrestrial and aquatic proxies indicate aridification in the Turkana Depression between 2.8 Ma and 2.5 Ma, the δ18Op of crocodilian teeth points to the stability of the local diversity of aquatic environments during this period."

Line 38: You introduce the study area, and that to me requires a map-figure. Consider making figure 2 (map) your figure 1, so that you can use that here.

Université de Poitiers - UFR SFA
Bât. B35 - TSA 51106 - 6, rue Michel Brunet - 86073 POITIERS Cedex 9 - France
http://palevoprim.labo.univ-poitiers.fr/ - axelle.gardin@univ-poitiers.fr

[Figure]

[Figure]

Laboratoire Paléontologie Evolution Paléoécosystèmes Paléoprimatologie
UMR 7262 CNRS – Université de Poitiers, France

**Axelle GARDIN**
Postdoctoral Researcher

We propose to combine the map of Africa with the diagram in Figure 1 to have a figure suitable for the whole introduction. Figure 2 will only consist of the map and log of the Shungura Fm.

Line 39-40: You introduce new work that challenges "previous models", but then you must explain what these previous models were. Please clarify or change the wording

We explain as follows: (e.g., level fluctuations of the paleolake that occupied the Turkana Depression in Nutz et al., 2020)

Line 50-52: This is a bit strong. It sounds like you are disqualifying the previous studies as unreliable, but I don't see immediately why that should be the case.

"However, so far, its freshwater environments have been only described by sedimentological studies and the analysis of invertebrate assemblages, despite their importance in capturing the interactions between climate, geodynamics, and aquatic communities (Van Damme et Gautier 1972; Peypouquet et al. 1979, 1980; Heinzelin et al. 1983; Van Boxclaer et Van Damme 2009). Therefore, contextual studies on ecosystems' aquatic components often lack reliable and complete paleoenvironment reconstructions."

We have carefully considered your comment and would like to clarify our intention. Our goal is in no way to discredit previous studies, but rather to highlight the reality of paleoenvironmental research. Aquatic environments have often been less favoured in past studies, mainly due to the terrestrial ecology of humans (key taxa of these contexts), which has led to a concentration on the terrestrial components of ecosystems. Moreover, we made an inventory and found that only 15% of studies on the Shungura Formation are interested in the aquatic component of ecosystems. Therefore, there is a lack of studies, tools, and methodologies suitable for describing aquatic environments with the same precision as terrestrial environments. This lack of focus on aquatic environments makes it difficult to obtain integrated, reliable, and complete paleoenvironmental reconstructions. We hope that this clarification explains our position and alleviates any impression of disparagement of previous studies.

We suggest rephrasing it as follows: Freshwater environments of the Shungura Fm. have primarily been explored through sedimentological studies and the analysis of invertebrate assemblages (e.g., Van Damme et Gautier 1972; Peypouquet et al. 1979, 1980; Heinzelin et al. 1983; Van Boxclaer et Van Damme 2009). However, to date, only about 15% of studies related to the Shungura Formation have addressed the aquatic component of its ecosystems, certainly mainly due to the terrestrial ecology of humans, which has led to a concentration on the terrestrial components of ecosystems. Nevertheless, it is essential to recognize the significance of aquatic environments in understanding the intricate interactions among climate, geodynamics, and aquatic communities. Therefore, there is a lack of studies, tools, and methodologies suitable for describing aquatic environments with the same precision as terrestrial environments. This scarcity of focus on aquatic environments in these contexts makes it difficult to obtain comprehensive paleoenvironmental reconstructions.

Line 76: you flag the "new" model here, but you don't explain what is new about it. Do that briefly, before you proceed with explaining the Shungura Fm.

"This paper proposes a new interpretative model of the $\delta 18Op$ of crocodilians by using the comprehensive knowledge of the physiology and ecology of crocodilians."

[Figure]

[Figure]

Laboratoire Paléontologie Evolution Paléoécosystèmes Paléoprimatologie
UMR 7262 CNRS – Université de Poitiers, France

**Axelle GARDIN**
Postdoctoral Researcher

We rephrase as follows: This paper proposes a new theoretical interpretative model of (1) the range of δ18Op values recorded in crocodilian teeth to describe the diversity of waterbodies accessible to crocodilians at a local scale, as well as (2) the evolution of minimum d18O values over time, by using the comprehensive knowledge of the physiology and ecology of crocodilians. This approach is adapted and applied to fossil teeth…

Line 83 – 85: I don't think I understand what you are saying here. Is the dating framework not good enough, or is the problem in the proxies? It may help to give some examples here, if you want to bring this point across.

"The growing knowledge of the evolution of its environments, the continuity of the series over several million years, and its accurate dating allow for discussing the new isotopic data in a well-constrained framework. However, uncertainties persist about the timing and factors inducing the main environmental Plio-Pleistocene changes, reported in a disparate way according to the proxies, the factors that control them and their impact on the evolution of fauna, including humans (e.g., Trauth et al., 2021)."

We rephrase as follow: "The growing knowledge of the evolution of its environments, the continuity of the series over several million years, and its precise dating allow for discussing the new isotopic data in a well-constrained framework. Because of disparate results across the various proxies, there remain uncertainties about the timing and factors that induced major environmental changes and how they affected the evolution of fauna, including humans in the Plio-Pleistocene in the Shungura Fm. and most widely in eastern Africa, (e.g., Trauth et al., 2021 for a recent review)."

Line 107-109: "The Shungura Fm recorded hydrological change over time: although it was dominated by a river system most of the time, the lake level rose until covering the site in the middle of Member G (2.06-1.95 Ma, Lorenyang highstand) and again at the top of Member L (1.19-1.09 Ma)" With "the Lake" you here mean the lake occupying the Turkana depression, right? Better to indicate that clearly for readers unfamiliar with the situation.

We rephrase as follows: The Shungura Fm recorded hydrological change over time: although it was dominated by a river system most of the time, the level of the paleolake occupying the Turkana Depression rose until covering the site in the middle of Member G (2.06-1.95 Ma, Lorenyang highstand) and again at the top of Member L (1.19-1.09 Ma) (Haesaerts et al., 1983; Nutz et al., 2020).

Line 124: "Rounded" is not what comes to mind for me when I look at these teeth. Is there another word possible (perhaps "stout" or something of the like)? We changed to stout instead of rounded in the revised manuscript.

Line 167-168: Based on 1SD statistics? please indicate that. Yes, we indicate it in the revised manuscript.

Line 170: instead of "control" better use "check for" or something of the like here. We corrected this.

Line 219-222: "The samples analyzed in this study lay on the linear relationship established for mammals and various northern African Cretaceous vertebrates (Iacumin et al., 1996; Amiot et

Université de Poitiers - UFR SFA
Bât. B35 - TSA 51106 - 6, rue Michel Brunet - 86073 POITIERS Cedex 9 - France
http://palevoprim.labo.univ-poitiers.fr/ - axelle.gardin@univ-poitiers.fr

al., 2010b)(adjusted R² = 0.50; Spearman's correlation coefficient = 0.74) (Fig. 5), suggesting preservation of their original oxygen isotope composition." See my comments on this higher up in this text. I don't understand how a deviation from the modern relationship demonstrates good preservation of your samples, and I don't see the point of comparing to (Cretaceous) fossils from elsewhere in this context? Please, refer to our detailed answer comment above.

Line 323: the word "enhanced" does not seem to fit here grammatically. We deleted this word.

Line 324: What is a close/shallow water system? I assume you are referring to systems that have experienced more evaporation, right? Here standing water (closed system) may be more important than it being shallow or not. Lake Turkana today, as a terminal lake, is not necessarily shallow, but still heavily 18O enriched due to evaporation.

This may lack clarity: We want to talk about a closed and/or shallow water system, that indeed will experience more evaporation and 18O enrichment. Both situations can occur and are described in the Shungura Formation by sedimentary analyses, so we prefer to keep this distinction (and clarify with "and/or").

Line 328: What is an "open water body" in this context? One could call Lake Turkana an open water body, but that is heavily evaporated. I think you are generally referring to the river setting (which has low water residence time and therewith relatively low evaporative loss)?

Here we are referring to hydrology rather than the environment. So "open water body" indicates as you say a water body with a short water residence time, low evaporation (e.g. exoreic for a lake), in opposition to pools or small lakes surrounded by wetlands. On the other hand, we use the term open lake (environment, as fig. 1) as a very wide lake, where the crocodile could swim for several kilometres without encountering a bank. Then one could say that today Lake Turkana is an open lake but in a closed basin.

We modify as follows: « The large range of δ18Op values in units B-10 to C-8 indicates that crocodilians occupied different water body types. Low δ18Op values suggest water bodies with low evaporation state (probably linked to short water residence time, through flowing, exoreic lake basin, river setting) or an 18O-depleted water source (e.g., unevaporated meteoric water from rivers or underground water); while the high values reflect shallow and/or isolated water bodies subject to strong evaporation (e.g., enclosed endoreic lake basin, pond, shallow small lake, swamp) or an 18O-enriched water source."

Line 339-340: "these data also point to stable local/regional conditions during this time interval" Please elaborate. What do you consider local/regional in the contrast between the Nachukui Fm and Shungura Fm that you just described?

We reformulate by incorporating the comment of the other referee: "These δ18Op values do not indicate aridification affecting freshwater environments between 2.8 Ma and 2.5 Ma (Members B and C). This contrasts with changes observed in the aquatic and terrestrial ecosystems of the eastern African Rift: such as lake level drop, landscape opening, and aridification (e.g., Maslin and Trauth, 2009; Trauth et al., 2009, 2021; Levin et al., 2011; Negash et al., 2015, 2020; Blondel et al., 2018, 2022; Nutz et al., 2020) (Fig. 8). Interestingly, the δ13C of soil carbonates increases, thus indicating aridification in the Shungura Fm but not in the Nachukui Fm in the West Turkana area (Levin et al., 2011; Levin, 2013; Nutz et al.,

2020) (Fig. 8). Nutz et al. explain (2020) that the δ13C record in the Shungura Fm reflects climatic conditions upstream on the Ethiopian Dome, and that of Nachukui Fm reflects more local and regional climatic conditions. The δ13C of the paleosols would therefore indicate a less significant supply of water from the Ethiopian Dome towards the Lower Omo Valley. On the other hand, the hydrological conditions remain stable in the western part of the Turkana Depression between 2.8 Ma and 2.5 Ma, thus possibly preserving a certain diversity of aquatic environments in the surroundings.".

Line 379: typo; "Moderb" We corrected this typo.

Line 388: "but could be due a rarefaction of rainfalls". Better to rephrase this: you mean that this could be due to changes in rainfall regime? We rephrase as follows: "but could be due a changes in rainfall regime (amount and/or origin of rainfall)".

**\*\*\*\*\***

Overall, we found that these suggestions were useful and certainly contributed to improving our manuscript. We very much look forward to seeing our paper published in the *Biogeosciences*.

We look forward to reading your reaction.

Sincerely,

Axelle Gardin

---

## Author Response (AR2)

Laboratoire Paléontologie Evolution Paléoécosystèmes Paléoprimatologie
UMR 7262 CNRS – Université de Poitiers, France

**Axelle GARDIN**
Postdoctoral Researcher

Orléans, November 28, 2023

Dear Editors,

Enclosed please find our revised manuscript bg-2023-125, entitled 'Stable oxygen isotopes of crocodilian tooth enamel allow tracking Plio-Pleistocene evolution of freshwater environments and climate in the Shungura Formation (Turkana Depression, Ethiopia)'.

We thank you for taking the time to review our revised manuscript and providing further feedback. We have incorporated most of the comments and suggestions proposed by the referees, and you will be able to follow the modifications and revised sections in the manuscript.

We were waiting for the referees' reports and the validation of our revised manuscript to send you a Graphical Abstract (Designed by EunJung Park at Science Graphic Design) which could possibly be used for the article published on your website.

In the paragraphs below, we provide answers to your comments:

**\*\*\*\*\***

Table 1: given the analytical uncertainty of the analyses, consider whether presenting d18O data with 2 decimals is appropriate. The new column (showing the range) has a mix of 1 and 2 decimals.

Thank you for this remark, we will only keep the first decimal place and make the necessary changes in Table 1.

Figure 4 and 5: increase font size for X and Y axis titles and tick mark labels so that they will remain clear in the final figure size.

We have made these changes.

Table 2, since you mention central Africa - note that there is also a reasonably long recent time series from a GNIP station in Kisangani (available through GNIP/WISER).

Thanks for suggesting it. However, the example of Goma in Central Africa was chosen because it is the station with the maximum precipitation (and not too far from eastern Africa), close to the estimates obtained if the variations in $\delta^{18}O$ were due to the amount of precipitation. The precipitation in Kisangani is much lower and would not serve for illustrating this example.

Figure 8: I realize there is a lot of info to show on this figure, but would suggest here also to try to increase the font sizes where possible to improve readability. The two panels on the right side show a gradient in the green color - the figure caption does not mention what this represents, does it merely change along the d13C and thus woody cover gradient (and hence, holds no extra information) ? Consider to either mention this explicitly, or remove the color gradient if not relevant.

We have made some modifications to "lighten" (slightly) the figure and enlarge the texts. Carbon isotope data for Shungura and Nachukui are shown in the same graph. We removed the color gradient that was not necessary and used the colors to distinguish Shungura and Nachukui.

Université de Poitiers - UFR SFA
Bât. B35 - TSA 51106 - 6, rue Michel Brunet - 86073 POITIERS Cedex 9 - France
http://palevoprim.labo.univ-poitiers.fr/ - axelle.gardin @univ-poitiers.fr

[Figure]

[Figure]

Laboratoire Paléontologie Evolution Paléoécosystèmes Paléoprimatologie
UMR 7262 CNRS – Université de Poitiers, France

**Axelle GARDIN**
Postdoctoral Researcher

✶✶✶✶✶

We very much look forward to seeing our paper published in the *Biogeosciences*.

Sincerely,

Axelle Gardin